# MAILEX: Email Event and Argument Extraction

**Saurabh Srivastava[†], Gaurav Singh[†] , Shou Matsumoto[†], Ali Raz[†],**
**Paulo Costa[†], Joshua Poore[#], Ziyu Yao[†]**
[†]George Mason University, [#]University of Maryland ARLIS
{ssrivas6, gsingh33, smatsum2, araz, pcosta, ziyuyao}@gmu.edu,
poorejc@umd.edu

## Abstract

In this work, we present the first dataset, MAILEX, for performing event extraction from conversational email threads. To this end, we first proposed a new taxonomy covering 10 event types and 76 arguments in the email domain. Our final dataset includes 1.5K email threads and ∼4K emails, which are annotated with totally ∼8K event instances. To understand the task challenges, we conducted a series of experiments comparing three types of approaches, i.e., fine-tuned sequence labeling, fine-tuned generative extraction, and few-shot in-context learning. Our results showed that the task of email event extraction is far from being addressed, due to challenges lying in, e.g., extracting non-continuous, shared trigger spans, extracting non-named entity arguments, and modeling the email conversational history. Our work thus suggests more future investigations in this domain-specific event extraction task.[1]

## 1 Introduction

Email has been one of the most widely used communication mediums, especially in professional and work environments. As the number of email users continues to rise, service providers are constantly looking for ways to improve user experience. With advancements in machine learning and natural language processing, email platforms have introduced a range of features aimed at helping users manage their inboxes and automate tasks (Feddern-Bekcan, 2008; Kannan et al., 2016; Chen et al., 2019), concurrent with research on request identification (Lampert et al., 2010), intent identification (Wang et al., 2019), meeting summarization (Deshmukh and Lee, 2022), task management (Zhang et al., 2022; Mukherjee et al., 2020), etc.

However, most of the existing work touches only one specific aspect of email information and thus

cannot connect with other relevant tasks. For instance, in Lampert et al. (2010) even after identifying emails containing requests, users will still need to manually search through the lengthy emails to find the actual request and then manage the tasks with separate tools. On the other hand, existing exploration on the email data can cover only a subset of potential events in email communications (e.g., requests or to-do tasks), whereas there are many others that also commonly happen and need proper management (e.g., delivery of data or information).

To facilitate more comprehensive downstream tasks on email data, in this paper, we introduce the task of event extraction on email threads. Event extraction (Grishman, 1997), or EE, is the task of extracting a specific occurrence of an event and its arguments. It is an important step for downstream tasks such as information retrieval, question answering, knowledge base population, etc. EE has long been studied in the context of news articles (Li et al., 2021; Yu et al., 2022; Du et al., 2022). As far as we know, there has not been such a dataset in the email domain. On the other hand, EE on email threads brings unique challenges such as performing information extraction in a conversational setting and needing to handle much longer and more verbose arguments, which cannot be studied with existing datasets.

To fill this gap, we first developed a taxonomy to describe events and arguments in email data. Specifically, we designed a set of 10 event classes and 76 arguments using the speech act theory of Cohen et al. (2004). Our event classes cover proposals, amendments, and deliveries of actionable events on meetings, actions, and data items (Section 2). Unlike existing EE datasets, each trigger in the email EE task is described with one Verb and one Noun Act (e.g., Deliver$_{Verb}$ Data$_{Noun}$), and the arguments are often long-span, non-named entities (e.g., a description of the meeting agenda), which make the task much more challenging. Based on

---

[1]The source code and dataset can be obtained from https://github.com/salokr/Email-Event-Extraction.

this taxonomy, we then proposed a new dataset, MAILEX, which consists of 1.5K email threads and ∼4K emails annotated with ∼8K events. The dataset achieves a substantial agreement between annotators.

Comparing three approaches, i.e., fine-tuned sequence labeling based on BERT (Devlin et al., 2018), fine-tuned generative EE based on BART (Lewis et al., 2019), and in-context learning using GPT-3.5,[2] we analyze the challenges in trigger and argument extraction. Our results highlight the need for advancements in handling non-continuous, shared triggers and long-span, non-named entity arguments while emphasizing the importance of effectively modeling email history. Moreover, the in-context learning of GPT-3.5 yields much worse performance, suggesting the challenge of this domain-specific task in the few-shot setting.

## 2 Taxonomy for Email Event Extraction

### 2.1 Verb and Noun Acts

In this work, we focus on extracting commonly seen events (e.g., scheduling meetings) from daily email communications. Our event definition follows the email speech act theory of Cohen et al. (2004). An email speech act describes the *sender* intent using a "verb-noun" pair, such as "Request$_{verb}$ Meeting$_{noun}$". As such, the email speech act carries the "actionable" information by the sender. In Cohen et al. (2004), a set of five verbs and four nouns are proposed (which could form 20 email speech acts). However, our preliminary study on email corpora (Minkov et al., 2008; Oard et al., 2015; Ulrich et al., 2008) reveals that most of them are not used frequently in daily communications (e.g., amending an action), or are not relevant to "events" (e.g., exchanging opinions). Therefore, we keep our focus on the most common 10 event types enabled by three verb acts (i.e., **Request**, **Deliver**, and **Amend**) and three noun acts (i.e., **Data**, **Meeting**, and **Action**). For the noun act "Data", we further consider three sub-categories: (a) **Meeting Data**, which refers to facts related to specific meetings (e.g., meeting date, location), (b) **Action Data**, which refers to facts related to a specific action or an activity (e.g., a deadline for approving a budget request, the person who approved the request, etc.), and (c) **Data** which refers to all other information irrelevant to meetings and actions, such as PDF files sent in emails. While this fine-grained noun

acts categorization may lead to skewed data distributions (Table 5), doing so allows us to easily connect the EE task with downstream applications. For example, when an EE model extracts meeting data, a downstream email reminder can be automatically set up to provide additional assistance, which will not be feasible if we simply merge all types of data information into one coarse category. Detailed descriptions of all the verb and noun acts can be found in Appendix A.1.

### 2.2 Event Types and Argument Roles

In total, we defined 10 event types with 76 argument roles, including a few "meta semantic roles" which come with pre-defined class spaces. We present three event types as examples below and show the full list in Appendix A.2. In the examples, we also denote the corresponding triggers (underlined) and [argument roles] (wrapped by "[ · ]").

**Request Data:** The event is triggered when the sender seeks data such as a file or a fact.

> Example: *Please send me [the summary of our meeting]$_{Data\ IdString}$* (Request Attribute: Data Value)

Here, "Data IdString" refers to the identity or description of the sender-requested data. We also introduce a meta semantic role "Request Attribute" to indicate the attribute that the sender queries from the data, which in practice is often the "Data Value", e.g., the specific PDF file of the meeting summary.

**Deliver Data:** The event is triggered when the sender provides or commits to provide certain data.

> Example: *Attached for your review [the summary of our meeting]$_{Data\ IdString}$.* (Confirmation: Positive)

For Deliver events, we introduce "Confirmation" (positive, negative, or tentative[3]) as a meta semantic role, affirming if the sender can provide the requested data information (i.e., when the noun act is *Data*), or acknowledge their attendance in meetings or participation in action events (i.e., when the noun act is *Meeting Data* or *Action Data*). Notably, the Confirmation role could be perceived as a form of "data" as well. In a conversational email setting, people often reply with brief responses such as "Sure" or "No, it doesn't work" when someone makes a request. By introducing the Confirmation

---

[2]https://platform.openai.com/docs/models.

[3]Rarely people may give uncertain responses such as "I'm not sure"; in that case, we mark it as "Unsure".

role, we can discern the sender's intent even though no concrete event information may be extracted from a short answer.

**Amend Data:** The event is triggered when the sender requests or indicates changes to a data record. In order to describe the type of change, we introduce a fixed set of "Amend Type" verbs including add, delete, and update. Additionally, we have observed that individuals frequently describe changes by providing context followed by the revision, as shown in the example below. Consequently, to differentiate between the various roles, we introduce two labels, "Context" and "Revision", and replace all argument roles of the Data act with two sets of copies for each (e.g., "Context: Data Type" and "Revision: Data Type" instead of the original "Data Type"). These modifications allow for more precise differentiation and description of the different aspects of the event and its roles.

> Example: *Can [you]$_{Members}$ change [the budget]$_{CNT:Data\ IdString}$ from [2K]$_{CNT:Data\ Value}$ to [3K]$_{REV:Data\ Value}$* (Amend Type: Update)

**Note on Non-Continuous, Shared Triggers.** Finally, we note that multiple events of the same type could be mentioned in one email. In that case, trigger words could be shared partially between events, which makes the dataset more challenging:

> Example: *Alice will approve the wire request and inform to Susan.*

In this example, two Deliver Action Data events share the trigger word "will".

## 3 The MAILEX Dataset

### 3.1 Data Annotation

We utilize the open-source Enron dataset (Minkov et al., 2008)[4] which comprises a collection of email data from 150 users. We considered the top 50 users with the highest inbox counts and randomly selected a range of 20 to 40 email threads for annotation. Note that all single-email threads have been removed in the interest of conversational event extraction. By focusing on a set of users, MAILEX could support personalization research, which we leave as future work. The annotation process involved annotators marking trigger words, event

---

[4]http://www-2.cs.cmu.edu/~enron/. Some prior work instead used Avacado (Oard et al., 2015); we did not choose it because it is not completely publicly available.

| Data Statistics | Total (train/dev/test) |
|---|---|
| # of email threads | 1,500 (1,200/150/150) |
| # of total emails | 3,936 (3,117/414/405) |
| # of non-event emails | 776 (636/70/70) |
| # of annotated events | 8,392 (6,571/946/875) |
| Avg. # of events of the same type appearing at least twice | 3.05 |
| Avg. # of words in an email | 64.400 |
| Avg. # of words in a trigger | 2.64 |
| Avg. # of words in an argument | 7.41 |

Table 1: MAILEX data statistics.

types, and argument roles for each email while considering the context of the email history. Two native English-speaking Computer Science students were recruited for the task and received 12 USD/hr for their work. To ensure accuracy, multiple rounds of training and discussions were conducted. Each email was annotated twice by each annotator, and annotations with agreement on event type, overlapping trigger words, and argument spans were retained. Specifically, for partially agreed triggers (but with agreement on the event type), we retained the overlapped word spans, and for partially agreed arguments (but similarly with agreement on the event type and having overlapped trigger spans), we similarly retain the overlapped word span. When two annotators did not agree on the event type or made no overlap in their annotated triggers, we abandoned the annotations completely; for more details and the annotation guideline, see Appendix B.1. In total, we collected a dataset consisting of 1.5K email threads, encompassing ~4K emails and ~8K events (Table 1).

**Inter-Annotator Agreement (IAA).** We measure two IAA values, one for triggers and their associated event types (i.e., whether annotators agree on the same trigger words and assign the same event type), and one for the argument roles (i.e., whether annotators agree on the argument role annotations for the same trigger and event type). For both calculations, we consider overlapping spans as indicating partial agreement and apply Cohen's kappa $\kappa$ (Cohen, 1960) at the word level while comparing the annotations. We obtained a $\kappa$ value of 0.791 (i.e., substantial agreement) for the trigger-event type IAA and 0.810 (i.e., almost perfect agreement) for the argument role IAA. For "meta semantic role" annotations, we did not observe disagreements between the annotators who had agreed on event triggers. We include analyses on the disagreement cases in Appendix B.2.

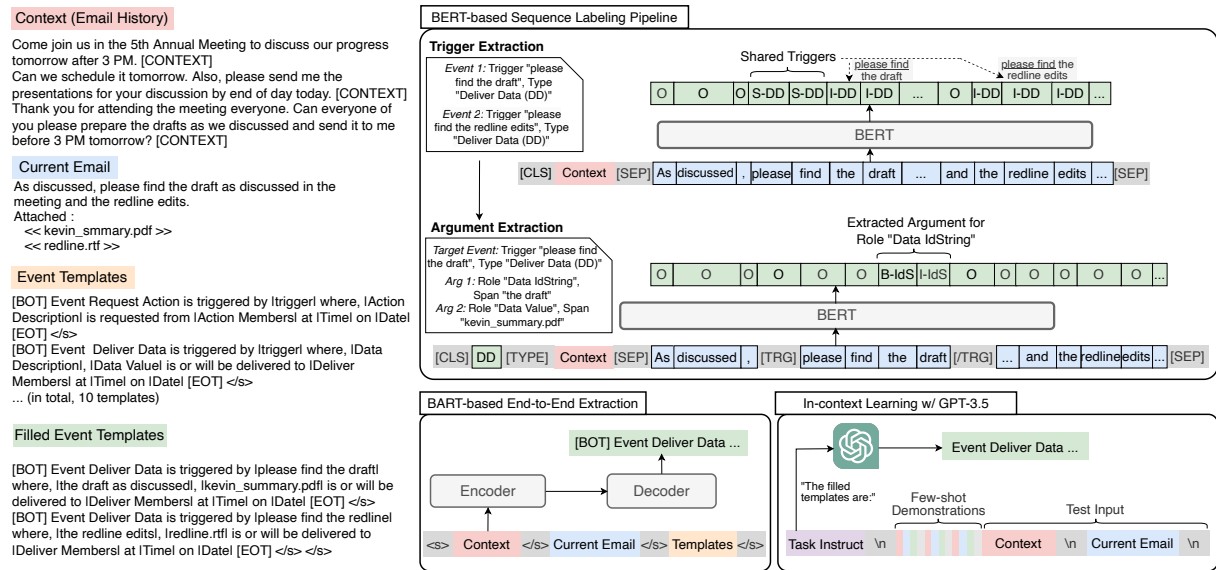

Figure 1: Illustrations of the three approaches we experimented with for email EE.

## 3.2 Data Statistic and Analysis

We present MAILEX statistics in Table 1. By looking into the details, MAILEX presents following unique characteristics and challenges:

**Imbalanced type and role distribution.** As shown in Table 5, the event distributions are imbalanced across different event types (e.g., events related to delivering data are more common than amendments); similarly for argument roles.

**Conversational context.** In a conversational setting, we observe common patterns between consecutive emails. For example, a request event is typically followed by a deliver or an amend event. Modeling the email context and capturing this intuition can thus be helpful for the task.

**Multiple events of the same types.** Unlike existing datasets, MAILEX often contains multiple instances of the same event type within a single email, such as multiple deliver data events. When such cases happen, on average the same event type recurs in ∼3 instances in the same email.

**Non-continuous, shared triggers.** Since MAILEX contains event classes with verb and noun acts, triggers signaling both acts may not necessarily be continuous, especially when they share spans, posing a new challenge for trigger identification.

**Non-named-entity arguments.** Argument spans for roles such as "Meeting Agenda" and "Action Description" may not necessarily be named entities; as an example, consider the "Meeting Agenda" argument in "*We will discuss the following items today [1] Actionable for this month . 2) Next month's*

*budget plan. . . . ]$_{Meeting\ Agenda}$.*" As a result, arguments in MAILEX can be much longer than conventional entity arguments and may even span over a few sentences. Unlike trigger spans, however, argument spans are always continuous spans.

**Non-event emails.** Some emails contain only non-event information, such as opinions and news information, and an intelligent EE model should not identify any events from them.

**Tabular data.** MAILEX also includes emails containing tabular data, which pose challenges due to their non-sentence-like sequential structure (see Figure 4 for example).

## 4 Methodology

### 4.1 Task Formulation

Each email thread $X = (X_1, \ldots, X_t, \ldots, X_T)$ consists of multiple emails, where $T$ is the total number of emails in a thread. Our goal is to extract events from the thread. This involves two sub-tasks: (1) **Trigger Extraction**, where we identify the trigger span within each email and determine the associated event type, and (2) **Argument Extraction**, where we identify spans within each email that serve as argument roles for the event instance. During event extraction for a given email $X_t$, only information before the current time step $t$ is used as context. This emulates a practical scenario where an intelligent system incrementally extracts events as new emails arrive. In this work, we explore three approaches to understand the task of email event extraction, as summarized in Figure 1.

## 4.2 BERT-based Sequence Labeling

Sequence labeling based on BIO tags is a classic approach to event extraction (Nguyen et al., 2016; Nguyen and Nguyen, 2018). In our work, we fine-tune two BERT models, one for trigger extraction and one for argument extraction, respectively.

For trigger extraction, in order to address the challenge of multiple events within a single email (Section 2.2), we additionally introduced a shared "S" tag. Each event type is assigned BIS tags, such as S/B/I-Request Meeting, while the tag O is used to denote non-trigger words common to all event types. Shared triggers among event instances of the same type are identified using S tags (see Figure 1 for an example). The input to BERT is organized as "[CLS] $X_1$ [CONTEXT] $X_2$. . . [CONTEXT] [SEP] $X_t$ [SEP]". Each word in the current email $X_t$ is then assigned a label from the set of BIOS tags based on its BERT representation.

For argument extraction, the BERT model is provided with a target trigger span and its event type. We similarly formulate this task as a BIO sequence labeling problem. However, unlike trigger extraction, arguments of the same event instance do not share spans. Therefore, we do not use any "S" tags in argument extraction. We prepare the input as the following to the BERT model: "[CLS] $type [TYPE] $X_1$ [CONTEXT] $X_2$ ... [CONTEXT] [SEP] $x_{t,1}$ ... [TRG] ... [/TRG] ... $x_{t,|X_t|}$ [SEP]". Here, $type is a placeholder for the event type. To encode the trigger span information, we introduce a pair of special tokens "[TRG]" and "[/TRG]" to indicate triggers in the current email $X_t$. In the case of non-continuous trigger spans, every segment of the trigger span will be wrapped by this pair of special tokens. The argument role label of each word is then predicted based on its BERT representation.

In addition, the argument extraction model also includes classifier heads for meta semantic roles prediction (Section 2.2), which will be jointly optimized in training. We refer readers to Appendix C.1 for details about the training and inference of the sequence labeling approach.

## 4.3 BART-based End-to-End Extraction

A drawback of sequence labeling approaches lies in that they cannot leverage the semantic meaning of the label names and thus may need massive annotations for effective generalization. Recent work has illustrated the promise of adopting pre-trained autoregressive language models for EE, where label names are explicitly spelled out during decoding (e.g., "*The meeting time is 7 AM*" for extracting the "Meeting Time" argument) and their semantics can thus be leveraged (Li et al., 2021; Du et al., 2022). Drawing inspiration from there, we design a set of event templates (see Figure 1 for an example and Appendix C.2 for all templates) and fine-tune a BART model to perform end-to-end EE.

For end-to-end extraction, the model's input comprises the email content and the template. Specifically, we prepare the input sequence as " $X_1$ [CONTEXT] $X_2$ ... [CONTEXT]  $X_t$  [BOT] $template_1 [EOT]  [BOT] $template_2 ... [EOT]  ", where "[BOT]" and "[EOT]" are special tokens indicating the template boundaries. With this setup, BART is trained to generate a sequence of templates, extracting the events from the email and their respective arguments. Importantly, the model only decodes templates for events present in the email, disregarding the ones for absent events. Moreover, in scenarios where an email contains multiple instances of the same event, the model produces multiple filled-out templates for each instance, all categorized under the same event type. All the generated templates are delimited via the special tokens "[BOT]" and "[EOT]".

## 4.4 In-context Learning with GPT-3.5

We further evaluate the performance of GPT-3.5 to understand if few-shot large language models have been able to perform well in closed-domain EE tasks. Similar to our BART-based EE model, we use GPT-3.5 for end-to-end extraction. Our prompt concatenates a task instruction, all the event templates, few-shot demonstrations, and the context and email body for the test example. We ensure the presence of all possible event types and arguments by carefully selecting K (K=5) shots of demonstrations from the training set. We present our prompt in Figure 8 in Appendix C.3. In experiments, we investigated both text-davinci-003 and gpt-3.5-turbo for a comparison.

## 5 Experiments

### 5.1 Experimental Setup

**Datasets.** We split MAILEX by email threads into training, development, and test sets with a ratio of 80, 10, and 10, ensuring that at least once each

| | Trigger | | | | | | Argument | | | | | |
| | Identification | | | Classification | | | Identification | | | Classification | | |
| | Precision | Recall | F1 | Precision | Recall | F1 | Precision | Recall | F1 | Precision | Recall | F1 |
|---|---|---|---|---|---|---|---|---|---|---|---|---|
| **BERT-based Sequence Labeling** | 0.581 | **0.499** | **0.537** | 0.566 | **0.486** | **0.523**[*] | 0.491 | **0.403** | **0.454** | 0.355 | **0.383** | **0.368** |
| w/ ground-truth triggers | - | - | - | - | - | - | 0.653 | 0.671 | 0.662 | 0.642 | 0.660 | 0.651 |
| w/o email thread history | 0.577 | 0.493 | 0.531 | 0.532 | 0.483 | 0.506 | 0.488 | 0.397 | 0.438 | 0.335 | 0.380 | 0.356 |
| **BART-based Generative Extraction** | **0.701** | 0.395 | 0.505 | **0.701** | 0.394 | 0.500 | **0.592** | 0.351 | 0.441 | **0.374** | 0.350 | 0.363 |
| w/ ground-truth triggers | - | - | - | - | - | - | 0.690 | 0.482 | 0.568 | 0.688 | 0.471 | 0.560 |
| w/o email thread history | 0.688 | 0.389 | 0.500 | 0.679 | 0.388 | 0.494 | 0.572 | 0.333 | 0.421 | 0.370 | 0.339 | 0.354 |
| **In-context Learning (GPT-3.5)** | | | | | | | | | | | | |
| text-davinci-003 | 0.167 | 0.171 | 0.169 | 0.100 | 0.101 | 0.100 | 0.068 | 0.069 | 0.068 | 0.058 | 0.060 | 0.058 |
| w/ ground-truth triggers | - | - | - | - | - | - | 0.379 | 0.356 | 0.367 | 0.349 | 0.330 | 0.338 |
| gpt-3.5-turbo | 0.183 | 0.095 | 0.121 | 0.098 | 0.060 | 0.072 | 0.058 | 0.045 | 0.050 | 0.056 | 0.040 | 0.048 |
| w/ ground-truth triggers | - | - | - | - | - | - | 0.256 | 0.198 | 0.223 | 0.242 | 0.190 | 0.211 |

Table 2: Results on MAILEX test set. For both fine-tuned and in-context learning, we additionally report their argument extraction performance when feeding ground-truth triggers ("w ground-truth trigger"). For the former, we also report their overall performance when the email thread history is ablated ("w/o email thread history"). [*] indicates significantly better performance than BART under a Wilcoxon signed-rank test (Wilcoxon, 1992) with a significance level $\alpha = 0.05$, whereas no significant difference was observed for Argument Classification F1.

event type is present in each of the three sets. The statistics for the sets are shown in Table 1.

**Evaluation Metrics.** We evaluate trigger and argument extraction using Precision, Recall, and F1 scores, following prior work (Du and Cardie, 2020; Sheng et al., 2021). For triggers, we consider a match when the identified span exactly matches the gold label (**Trigger Identification**) and is correctly classified into the event type (**Trigger Classification**). For arguments, we assess **Argument Identification** (i.e., whether an argument span is correctly identified) and **Argument Classification** (i.e., whether the argument span is additionally correctly classified into the true role). Unlike trigger evaluation, partial matching is allowed for arguments to encourage more fair comparison, especially for non-named entity arguments with long spans. This aligns with similar evaluation strategies used by Li et al. (2021). Finally, we note that the argument evaluation reports an end-to-end extraction performance; for BERT-based sequence labeling, only the model-extracted triggers are fed for argument extraction during evaluation. More implementation details are provided in Appendix D.

## 5.2 Experimental Results and Analyses

### 5.2.1 Main Results

Table 2 shows the model performance. We observe that the BERT-based sequence labeling pipeline and the BART-based approach achieve comparable end-to-end argument classification performance, though the former outperforms the latter in trigger extraction. On the other hand, BART exhibits high precision in making decisions, yet struggles in recall. A qualitative examination of the dev set sug-

gests that BART occasionally fails to copy from the current email, which leads to low recall. Moreover, for trigger identification and classification, BART achieves close F1s, suggesting that once it identifies the span, it accurately classifies the trigger.

Finally, we note much worse overall performance by both versions of GPT-3.5 in-context learning, which we will carefully discuss in Section 5.2.5. In light of this underwhelming performance, our subsequent analyses will mostly focus on the two fine-tuned approaches.

### 5.2.2 Challenges in Extracting Triggers

**Identifying Minimal, Complete Trigger Spans.** Our annotation guidelines (Appendix B.1) constrains a trigger to be a minimal sequence of words or phrases triggering an event. We observed that both models fail to adhere to this constraint and make mistakes by adding additional trivial details, e.g., for an email with the ground-truth trigger "*Will meet*", BERT predicted "*Will meet upstairs*".

Additionally, we noticed a few instances where both models fail to identify the complete trigger span, resulting in propagated errors in trigger classification. For example, for an email with the ground-truth trigger "*amended the deal*", BERT predicted a partial trigger "*amended*". It is worth emphasizing that in the ground-truth trigger, the phrase "the deal" informs the recipient about an amendment made on the data "*the deal*", thereby triggering the Amend Data event. Failure to identify the complete trigger span incorrectly triggered a Deliver Action Data event.

**Classifying the Noun Acts of Triggers.** In trigger classification, models struggle to properly classify

noun acts associated with triggers. For example, we observed instances where the true event Request Action Data was confused with Request Action 39% of the time, and Deliver Meeting Data was confused with Deliver Action Data 27% of the time (Figure 6 in Appendix E.1). Such challenges arise from similar words or phrases used by email senders to trigger specific noun acts. For instance, when the trigger is "*will attend the seminar*" BERT fails to recognize that a seminar is a type of meeting, resulting in incorrect classification as Deliver Action Data instead of Deliver Meeting Data. This highlights a challenge in MAILEX, where abstract event-type definitions lead to language variations and variations in noun acts of triggers. On the contrary, previous event extraction datasets (Walker et al., 2006; Wang et al., 2020) have focused mainly on verb-act triggers, overlooking the complexities of noun acts and the resulting language variations.

**Extracting Non-Continuous, Shared Triggers.** We observe that BERT was unable to segment spans that trigger multiple events. For example, in emails containing shared triggers with short distances, such as "*Attached the report and the redlines.*", it identifies one Deliver Data event with the trigger "*Attached the report and the redlines*". Meanwhile, when there is a long distance between the two partially shared triggers, BERT can identify only the first one. We include examples in Appendix E.2. Intriguingly, BART was able to correctly extract shared triggers with shorter distances in the majority of cases, though it still couldn't handle the longer distances. These findings are consistent with a similar study conducted by Sheng et al. (2021) where the authors also argue the limitations of sequence labeling approaches for such shared triggers.

### 5.2.3 Challenges in Extracting Arguments

**Error Propagation from Trigger Extraction.** In Table 2, we present each model's performance on argument extraction when ground-truth triggers are fed, so as to understand whether the low end-to-end argument extraction performance of the two fine-tuned models is caused by error propagated by trigger extraction. We note that even with the gold triggers, both models still fall short of achieving perfect argument extraction results, highlighting the challenging nature of both extraction tasks. Moreover, with ground-truth triggers, the sequence labeling pipeline outperforms BART by around

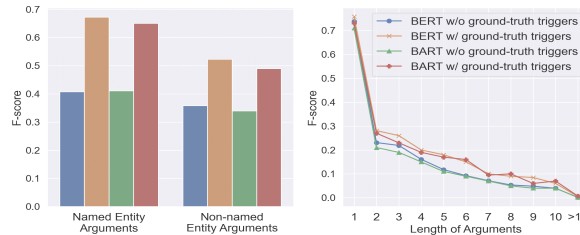

Figure 2: Argument classification results on MAILEX dev set, categorized by whether the argument is a named entity (left) and by its length (right). For spans of length more than 10 we show macro-average of their F1s. All the models struggle to correctly extract non-named entities, long-span arguments.

9% Argument Classification F1. This implies a stronger argument extraction performance from the former model. In our conjecture, this can be attributed to the fact that the pipeline approach has an independently learned argument extraction model, while the BART approach has to learn both extraction tasks within the same model.

**Extracting Non-Named Entity Arguments.** In Figure 2, we break down each model's argument extraction performance by named vs. non-named entity argument as well as the argument length. The results indicate that all models struggle to extract non-named entity arguments, particularly those with longer spans. This observation thus implies the need for more advanced modeling strategies in future research.

### 5.2.4 Importance of Modeling Email History

In Table 2, we present model performance when ablating the modeling of the email history (i.e., "context" in Figure 1). As expected, we observed performance drops for both BERT and BART in all metrics. This emphasizes the importance of modeling the conversational history in the email thread. To corroborate this, we conducted a study of randomly sampled 50 emails and found that 11 (22%) emails required the previous turn in event decision-making. We note that this percentage is much larger than the observed performance drop. We attribute this inconsistency to the ineffective modeling of email history when our approaches simply concatenate all the prior email bodies. This thus calls for future exploration, such as selectively including prior emails only when they are helpful for EE from the current email.

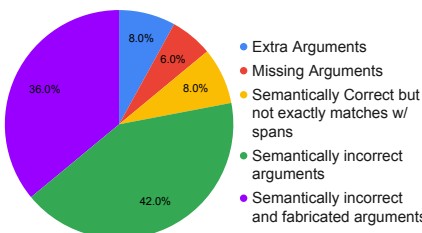

Figure 3: Distribution of erroneous arguments extracted by `gpt-3.5-turbo`.

### 5.2.5 Analysis of In-context Learning

Both versions of GPT-3.5 in-context learning behaved substantially worse (e.g., 0.058 and 0.048 Argument Classification F1 in end-to-end evaluation) compared to the fine-tuned approaches. To understand whether the challenge lies solely in extracting triggers, in Table 2, we similarly present results with ground-truth triggers as how we analyzed the fine-tuned models in Section 5.2.3. However, the results show that even with ground-truth triggers, the few-shot argument extraction is still very challenging (more than 0.3 Arg. Class. F1 behind the fine-tuned models).

We analyzed 50 randomly sampled erroneous predictions by `gpt-3.5-turbo` w/ gold triggers, and categorized errors in its extracted argument values in Figure 3. The most common mistakes made by the models include semantically incorrect arguments such as extracting an incorrect person as the meeting member (42%). In this case, the incorrect arguments are still valid entities mentioned in the email. However, another common mistake (36%) is generating not only semantically incorrect but also fabricated, non-existing entities in the email as arguments. Approximately 8% of the generated arguments are semantically correct but not exact spans copied from the email, such as a summarized version of the meeting agenda. Other error types include introducing extra arguments (8%) or missing arguments (6%); for the former, the model assigns the sender as an extra member in all the failure cases. We include examples in Table 11. In addition, `gpt-3.5-turbo` also made errors when generating unused argument placeholders of the event templates, which we discuss in Appendix E.3. Notably, `text-davinci-003` rarely generates fabricated arguments, and it obtains better performance particularly because it made much fewer mistakes when generating argument placeholders.

We also note that due to the word limit imposed by GPT-3.5, for some test examples, we have to prune the email thread input, which could lead to a loss of information. Designing prompts that allow large language models to ground to long context is thus an important future direction.

## 6 Related Work

**Event Extraction Models.** Earlier work on EE tasks has typically followed a pipeline approach to identify triggers before extracting arguments (Ji and Grishman, 2008; Liao and Grishman, 2010; Du and Cardie, 2020). Alternatively, joint sequence labeling approaches (Nguyen et al., 2016; Nguyen and Nguyen, 2018) perform trigger extraction and argument extraction simultaneously, employing a unified decoder that tags the sentence in a single pass. A recent trend formulates EE as an extractive question-answering problem (Du and Cardie, 2020; Liu et al., 2020) which induces the language knowledge from pre-trained language models by converting EE tasks to reading comprehension tasks via a question template. With the help of pre-trained encoder-decoder Transformer architectures such as BART and T5 (Raffel et al., 2020), there is also some recent work converting extraction tasks to generation tasks (Li et al., 2021; Lu et al., 2021). Finally, prompt-tuning (Dai et al., 2022; Ma et al., 2022) and few-shot in-context learning (Gao et al., 2023) have emerged as promising solutions to combat the "low resources" constraint of EE.

In this work, we experimented with three approaches, i.e., a pipeline of sequence labeling, the BART-based generative extraction, and few-shot in-context learning using GPT-3.5. Particularly for sequence labeling, we introduced a "S" tag to handle shared triggers. Our experiments compared these approaches and shed light on future research on email EE.

**Event Extraction Datasets.** The Automatic Content Extraction, ACE05, dataset (Walker et al., 2006) has been the standard evaluation benchmark for EE. Similar to ours, there are also datasets focused on specific domains, such as drug safety (Sun et al., 2022), news headlines (Deng et al., 2022), and business and financial domain (Capet et al., 2008). While most existing EE datasets aim to extract information from individual sentences, several attempts have been made to extend the extraction task to multiple sentences (Ebner et al., 2020) or documents (Eirew et al., 2022). As far as we know, MAILEX is the first comprehensive dataset for EE in the email domain. As discussed in Section 3.2,

it brings multiple unique challenges such as the conversational context and the need to model non-named entity arguments, which were not covered by prior datasets.

**Other NLP research on Email Data.** Previous research on emails can be categorized into keyword and action extraction (Turney, 2000), request identification (Lampert et al., 2010), modeling action items in emails (Lin et al., 2018), subject line generation (Xue et al., 2020), to-do generation (Mukherjee et al., 2020), and text summarization (Deshmukh and Lee, 2022). There has also been considerable research on identifying speech acts or tasks in emails (Cohen et al., 2004; Carvalho and Cohen, 2005) and how it can be robustly adapted across diverse email corpora (Azarbonyad et al., 2019). Recent work on task management automatically extracts actionable items from emails, generates faithful to-do items, and then aligns them to the correct users (Zhang et al., 2022). MAILEX unifies the majority of these tasks (such as handling requests, creating to-dos, etc) and covers a wide range of events in email communications.

# 7 Conclusion

In this paper, we have proposed a new task of extracting events and their arguments from conversational email data. To motivate future research in this direction, we also present a new dataset MAILEX, including a new taxonomy to describe common events mentioned in emails. We also conduct a series of evaluations on MAILEX, concluding that email EE is far from being addressed and more advanced methodologies are needed.

## Limitations

While we aim to advocate the new task of EE in the email domain, our approaches can be significantly improved in the future. For example, as pointed out in Section 5.2.4, modeling email history is crucial for more accurate EE in a conversational setting. While we directly concatenate all the previous emails to extract events from the current turn, future work can design more specialized architectures for it such as applying an attention mechanism to retrieve only the relevant emails from the history. One could also use the dynamic memory similar to that of Du et al. (2022) and store only the extracted events (as opposed to the raw texts) from the email history. In addition, future work can further advance our approaches by modeling the sequential

event constraints (e.g., amendments often follow the proposal of an event), as well as proposing better modeling strategies to handle the long-text, non-named entity arguments in emails. Finally, it could be worth investigating the application of open-source Chat Language Models (e.g., Vicuna (Chiang et al., 2023), FastChat (Zheng et al., 2023), and Koala (Geng et al., 2023)) in this conversational EE task.

Another limitation of our work lies in the limited contexts of the Enron dataset, which is the source corpus of our annotations. As emails in the Enron dataset are all conversations among Enron employees or between Enron employees and outsiders, the resulting MAILEX still retains this context footprint and is not a fully open-domain one. However, despite this constraint, our taxonomy of email EE is not limited to only business contexts. As highlighted in Section 2, our taxonomy, inspired by Cohen et al. (2004), is tailored for task-oriented email communications, with the goal of extracting "actionable" items conveyed by the sender. While the majority of the MAILEX focuses on business-related dialogues, it also touches down the realm of informal and personal communications. Such emails might delve into personal work reflections or family-related job discussions. This diversity is consistent with the findings of Alkhereyf and Rambow (2017), which revealed a substantial volume of personal communications in the Enron collection. Given that the Enron dataset is, to our knowledge, the only comprehensive and publicly available email corpus, MAILEX offers invaluable potential for subsequent email EE research, despite its specific contextual nature.

## Ethical Statements

Our annotations are based on a fully open-source dataset (Enron), and our developed models will be open-source as well. We expect that our work can have a strong broader impact. For example, our dataset and the developed models can be used to enable more advanced personal assistants based on daily emails, which can improve workplace productivity or help people with difficulty in reading and processing a large volume of emails. However, given that even the best-performing EE models in our experiments cannot precisely extract the stated information and may even fabricate contents, additional verification tools and proper user guidance will be needed, although we anticipate that the ex-

traction performance can be significantly improved in the future.

## Acknowledgements

This work was supported by the United States Government under contract FA8702-15-D-0002, via subcontract through the University of Maryland. The views, opinions, and/or filings contained in this material are those of the author(s) and should not be construed as an official position, policy, or decision of the Government of the United States or Carnegie Mellon University or the Software Engineering Institute unless designated by other documentation. This project was also supported by resources provided by the Office of Research Computing at George Mason University (https://orc.gmu.edu) and funded in part by grants from the National Science Foundation (Awards Number 1625039 and 2018631). Finally, Saurabh and Ziyu appreciate the funding support from George Mason College of Engineering and Computing.

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

## A Taxonomy Details

### A.1 Verb and Noun Acts

We consider three *verb acts*. When introducing these acts, we also indicate the corresponding *triggers (underlined)* that signal their presence. (1) **Request**: The request act is triggered when the sender intends to perform an act of asking or ordering something formally or informally. Example: *Can you please send me a summary of our meeting yesterday?* (2) **Deliver**: The deliver act provides or *commits to provide* something, such as a file, an answer to a query, or information about events (e.g., the location of a meeting). Example: *I will send you the summary report of our meeting.* (3) **Amend**: An amend act requests or informs about a change in some earlier proposals, e.g., to change the meeting date or contact information in a database. Example: *Can you please update the summary report of our meeting?*

We also define three *noun acts*, which describe the event entities. (1) **Data**: Data can be a piece of information, such as a concrete file or an abstract fact. It is typically defined with an "IdString"(e.g., "the summary report of our meeting yesterday") and a "Value" (e.g., an attached PDF file). The fact includes event-relevant information such as the **Meeting Data** (e.g., the date when a meeting will be held) and the **Action Data** (e.g., the address where a package should be mailed). As we focus on actionable events, we do not consider subjective information (e.g., opinions) or objective information that is too complicated to be framed as data (e.g., news information), but would cover very light "is-a" facts (e.g., Skilling is the CEO of Enron, where the data IdString is "CEO of Enron" and the Value is "Skilling"). (2) **Meeting**: We define a meeting as a gathering of people for a discussion to achieve a common goal or for entertainment. We also consider a phone call or trip as a meeting. (3) **Action**: An action refers to an activity that has to be done or will be done, such as signing a document or sending a mail package. Note that the "activity" here does not include "meeting", which has been covered by the previous noun act. Similar to verb acts, each noun acts will be signaled by a certain trigger, as to be illustrated in the next section.

### A.2 Complete Event and Argument Role Definitions

We now present all 10 event types and their respective argument roles (wrapped within "[ · ]" in examples). In total, they result in 76 argument roles at the event level by combing the roles from the verb and the noun act for each event type (e.g., for Request Data event, there are 8 argument roles including Request Members, Request Date, Request Time, Request Attribute, Data Type, Data IdString, Data Value, and Data Owner). We also introduce several **"meta semantic roles"** with pre-defined

| Act | | Arguments |
|-----|---|-----------|
| Request | Members | The recipient of the request event. |
| | Date | The date when the event will (or has) happen (or happened). |
| | Time | The time when the event will (or has) happen (or happened). |
| | Attribute | The attribute requested for the noun event such as data, meeting time, etc. |
| Deliver | Members | The recipient of the corresponding event. |
| | Date | The date when the event will (or has) happen (or happened). |
| | Time | The time when the event will (or has) happen (or happened). |
| | Confirmation | Acknowledgment of the sender (positive/negative/unsure) |
| Amend | Type | The amend operation (add/delete/update) |
| | Members | The person to oversee the amend event. |
| | Time | The time when the amend action happens. |
| | Date | The date when the amend action happens. |
| Data | Type | The type of data (e.g., PDF, is-a facts). |
| | IdString | A string describing or identifying the data. |
| | Value | The actual data file or fact. |
| | Owner | The person to whom the data belongs. |
| Meeting | Members | Attendees of the meeting. |
| | Agenda | The topic of discussion for the meeting. |
| | Name | A reference name for the meeting. |
| | Location | The (physical or virtual) place where the meeting will be held. |
| | Date | The date on which the meeting will (or has) happen (or happened). |
| | Time | The time at which the meeting will (or has) happen (or happened). |
| Action | Members | Attendees of the activity. |
| | Description | A summary of the action. |
| | Date | The date on which the activity will (or has) happen (or happened). |
| | Time | The time at which the activity will (or has) happen (or happened). |

Table 3: Descriptions of argument roles for Verb (upper) and Noun (bottom) acts.

class spaces for some event types. The complete argument role definitions for each Verb or Noun Act can be found in Table 3, and the list of 76 argument roles can be found in Table 4.

Triggers of each event (for both Verb Act and Noun Act) are underlined. Note that an event trigger could span over non-continuous words since people may not necessarily describe verbs and nouns consecutively. This also allows us to keep the trigger words as concise as possible rather than marking a continuous but much longer text span.

**Request Data:** The event is triggered when the sender seeks data such as a file or a fact.

> Example 1: *Please send me [the summary of our meeting]$_{Data\ IdString}$* (Request Attribute: Data Value);
> Example 2: *Who owns [the survey report]$_{Data\ IdString}$?* (Request Attribute: Data Owner)

For Request Data/Meeting Data/Action Data, we introduce a meta semantic role "Request Attribute" to indicate the attribute that the sender queries from the data. In practice, we consider four data attributes: Type, IdString, Value, and Owner.

**Deliver Data:** The event is triggered when the sender provides or commits to provide certain data.

> Example 1: *Attached for your review [the summary of our meeting]$_{Data\ IdString}$.* (Confirmation: Positive);
> Example 2: *I don't have [that]$_{Data\ IdString}$.* (Confirmation: Negative)

For Deliver events, we introduce "Confirmation" (positive, negative, or tentative as a meta semantic role, affirming if the sender can provide the requested data information (when the Noun Act is *Data*), or acknowledge their attendance in meetings or participation in action events (when the Noun Act is *Meeting Data* or *Action Data*). Notably, the Confirmation role could be perceived as a form of "data" as well. In a conversational email setting, people often reply with brief responses such as "Sure" or "No, it doesn't work" when someone makes a request. By introducing the Confirmation role, we can discern the sender's intent even though no concrete event information may be extracted from a short answer.

**Amend Data:** The event is triggered when the sender requests or indicates changes to a data record. In order to describe the type of change, we introduce a fixed set of "Amend Type" verbs including add, delete, or update. Additionally, we have observed that individuals frequently describe changes by providing context followed by the revision, as shown in Example 1. Consequently, to differentiate between the various roles, we introduce two labels, "Context" and "Revision", and replace all four argument roles (Table 3) for Data act with two sets of copies for each (e.g., "Context: Data Type" and "Revision: Data Type" instead of the original "Data Type"). These modifications allow for more precise differentiation and description of the different aspects of the event and its roles.

> Example 1: *Can [you]$_{Members}$ change [the budget]$_{CNT:Data\ IdString}$ from [2K]$_{CNT:Data\ Value}$ to [3K]$_{REV:Data\ Value}$* (Amend Type: Update);
> Example 2: *Can you please update [the summary report]$_{CNT:Data\ IdString}$ of our meeting?* (Amend Type: Update).

| Request Events | | | | |
|---|---|---|---|---|
| **Request Meeting** | | **Request Data** | | **Request Action** |
| Meeting Members | Meeting Location | Request Date | Request members | Action Date |
| Meeting Agenda | Meeting Date | Data IdString | Data Owner | Action Members |
| Meeting Name | | Request Time | Data Type | Action Description |
| Meeting Time | | Requested Attribute | | Action Time |
| **Request Action Data** | | **Request Meeting Data** | | |
| Context: Action Time | Context: Request Members | Context: Meeting Date | Context: Meeting Time | |
| Context: Action Members | Context: Action Date | Context: Meeting Agenda | Context: Meeting Members | |
| Context: Action Description | | Context: Request Members | Context: Meeting Location | |
| Requested Attribute | | Context: Meeting Name | Requested Attribute | |

| Deliver Events | | | | | |
|---|---|---|---|---|---|
| **Deliver Data** | | **Deliver Action Data** | | **Deliver Meeting Data** | |
| Deliver Members | Data IdString | Action Date | Action Time | Meeting Members | Meeting Time |
| Data Value | Deliver Time | Action Members | | Meeting Name | Meeting Date |
| Deliver Date | Data Type | Action Description | | Meeting Agenda | Meeting Location |
| Deliver Confirmation | | Deliver Confirmation | | Deliver Confirmation | |

| Amend Events | | | | | |
|---|---|---|---|---|---|
| **Amend Data** | | **Amend Meeting Data** | | | |
| Context: Data Type | Context: Amend Date | Context: Meeting Members | Context: Meeting Name | Revision: Meeting Date | Context: Amend Time |
| Revision: Data Type | Context: Amend Time | Revision: Meeting Members | Context: Meeting Time | Context: Meeting Location | Revision: Amend Time |
| Context: Data Value | Amend Type | Context: Meeting Agenda | Revision: Meeting Location | Revision: Meeting Time | Amend Type |
| Revision: Data Value | | Revision: Meeting Agenda | Context: Meeting Date | Context: Amend Date | |
| Context: Amend Members | | Context: Amend Members | Revision: Amend Members | Revision: Amend Date | |

Table 4: All event arguments in MAILEX. We remove arguments that are trivial (such as Deliver Members for events Deliver Action Data and Deliver Meeting Data) or are not frequent (such as Data Owner in Amend Data event). In total, we keep 76 argument roles in the final version of MAILEX.

**Request Meeting:** The event is triggered when the sender proposes a meeting.

Example: *[Alice]$_{Meeting\ Members}$ has proposed a meeting on [Tuesday]$_{Meeting\ Date}$.*

**Request Meeting Data:** The sender triggers the event by requesting information about a certain meeting. By using the "Request Attribute" for the event, the sender could request one or more meeting attributes such as the Date and Time of the meeting.

Example: *Where is [the meeting]$_{Meeting\ Name}$ on [Tuesday]$_{Meeting\ Date}$?* (Request Attribute: Meeting Location)

**Deliver Meeting Data:** The event is triggered when the sender provides information about a certain meeting. The sender can acknowledge the presence of both the sender and any other attendees using the "Confirmation" attribute.

Example: *[Alice]$_{Members}$ will attend the [Tuesday]$_{Date}$ [Board meeting]$_{Meeting\ Name}$.* (Confirmation: Positive)

**Amend Meeting Data:** The event is triggered when the sender requests or informs of changes to an already proposed meeting event.

Example: *Can we reschedule [the meeting]$_{CNT:Meeting\ Name}$ on [Tuesday]$_{CNT:Meeting\ Date}$*

*to [Friday]$_{REV:Meeting\ Date}$ instead?* (Amend Type: Update)

**Request Action:** The event is triggered when the sender proposes an activity or an action (e.g., playing a sport, signing a document, etc.).

Example: *Please [approve Alice's travel request]$_{Action\ Description}$.*

**Request Action Data:** The event is triggered when the sender seeks information about an action event.

Example: *Who [approved the travel request]$_{Action\ Description}$?* (Request Attribute: Action Members)

**Deliver Action Data:** The event is triggered when the sender provides information about an action event. The "Confirmation" attribute serves the purpose of acknowledging the presence of the sender and any other individuals involved in the event.

Example 1: *[John]$_{Action\ Members}$ [approved the travel request]$_{Action\ Description}$* (Confirmation: Positive);
Example 2: *[Alice]$_{Action\ Members}$ has agreed to [deliver mail]$_{Action\ Description}$.* (Confirmation: Positive)

# B MAILEX Dataset Details

## B.1 Annotation Details and Guidelines

MAILEX annotations were done in multiple rounds due to the challenges discussed in Section 3.1. For

| Event Type | % | Frequent Argument Roles |
|---|---|---|
| Request Data | 7.700 | Data IdString (72%), Request Members (23%), Request Date (2%) |
| Request Action | 14.985 | Action Description (54%), Action Members (35%), Action Date (6%) |
| Request Meeting | 2.775 | Meeting Members (31%), Meeting Agenda (21%), Meeting Date (18%) |
| Request Action Data | 2.456 | Action Description (51%), Action Members (38%), Request Members (8%) |
| Request Meeting Data | 0.541 | Meeting Members (31%), Meeting Agenda (21%), Meeting Date (18%) |
| Deliver Data | 20.452 | Data IdString (48%), Data Value (39%), Deliver Members (10%) |
| Deliver Action Data | 34.439 | Action Description (46%), Action Members (41%), Action Date (9%) |
| Deliver Meeting Data | 5.176 | Meeting Members (34%), Meeting Date (19%), Meeting Time (12%) |
| Amend Data | 2.054 | Amend Members (26%), (Context) Data IdString (25%), (Revision) Data Value (25%) |
| Amend Meeting Data | 0.569 | (Revision) Meeting Time (22%), (Revision) Meeting Date (19%), (Context) Meeting Name (16%) |

Table 5: Distributions of event types (in percentage) and frequent argument roles in MAILEX. We have not included the rare events Amend Action Data (0.028%) and Non-Event annotations "O" (8.825) in the table.

consistent annotations, annotators were instructed to annotate one email at a time, considering the email history as context (see Figure 5 for the annotation interface). Each email could have multiple events, and annotators marked trigger words, event types, and argument roles. For trigger words, annotators indicated the minimal span of words in the email that triggered an event. Event types were selected from pre-defined labels. Argument roles were annotated using the BIO format, with annotators marking the beginning (B) and inside (I) spans of the arguments while leaving non-arguments outside (O). For Amend events, "Context" and "Revision" were included in the BI tags (e.g., "B-CNT:Meeting Date" or "B-REV:Meeting Date"). Annotators also assigned pre-defined labels for meta semantic roles from pre-defined labels accordingly.

Two native English-speaking Computer Science students were recruited for the annotation task and were paid 12 USD per hour. Multiple rounds of training and discussions were conducted to ensure an understanding of events and arguments. Each email was annotated twice by each annotator, and we retained event annotations with agreement on

event type, overlapping trigger words, and overlapping argument spans for the same role. Probing into the annotations, we found that the non-overlapping partial text spans are typically trivial words such as an article "the". We use Jaccard similarity larger than .3 as the "overlapping" criterion. Threads with a total disagreement on event triggers and arguments were discarded. In total, we obtain 1,500 email threads covering ∼4K emails and ∼8K events.

## B.2 Examples of Partially Agreed and Disagreed Annotations

In practice, most partially agreed annotations happen when annotators inconsistently marked trivial words (e.g., an article "the") or referred to the same entity mentioned with different details (e.g., "Attached agreement report" and "Attached report"), while they agree on the actual trigger or argument concepts. This gives us a $\kappa$ value of 0.791 (i.e., substantial agreement) for the trigger-event type IAA and 0.810 (i.e., almost perfect agreement) for the argument role IAA.

We sampled a few annotations with total or partial disagreement and manually analyze them. In most cases, the total disagreement was caused by task complexity and language ambiguity. For example, in one email, the sender informed the recipient of a "to-do list" to which one annotator marked it as a Deliver Data event since the sender delivered a list of the informative items, while the other annotator considered it a Request Action since the sender had instructed a list of actions. Such disagreed annotations have been removed from our dataset. For partially disagreed cases, we often observed disagreement on trivial words, as discussed in IAA calculation (Section 3.1). We present more examples in Table 6.

## B.3 Dataset Analysis

**Event Types and Argument Roles Distribution.** In Table 5, we present the distribution of event types and argument roles in MAILEX. We observe that events related to deliver acts are more frequent than others and argument roles such as Members, Descriptions, and IdString are more frequent than Date and Time. **Tabular Data in Email Text.** As mentioned in Section 3.2, MAILEX could contains emails which have non-sequential sentence structure such as Tables. Figure 4, we present an example table from MAILEX. For the sake of sim-

```
FROM: Winckowski, Michele
TO: Blair, Lynn; Bodnar, Michael
SUBJECT: SBA Contracts
|
Contract     Shippers                    Contract Date
107018       Tenaska Marketing Ventures   Nov 1 2000
107019       Texaco Gas Marketing         Nov 1 2000
107021       OGE Energy Resources         Nov 1 2000
107989       Tenaska Marketing            Jun 1 2001
108021       Arkla Energy Marketing Co.   Jul 1 2001
108284       Tensaka Marketing Ventures   Nov 1 2001
108290       Texaco Gas Marketing         Nov 1 2001
108281       El Paso Merchant Energy.     Nov 1 2001
108282       UtiliCorp United, Inc.       Nov 1 2001
108283       Engage Energy American Nov 1 2001
These are the SBA contracts that were provided during the due diligence.  I'd like to be
able to verify and support that the changes in the business requirements for the
scheduling priority for these SBA contracts do not result in an impact to NNG's primary
firm shippers.  Your assistance would be greatly appreciated. Thanks MW
________________________________
```

Figure 4: Example table from MAILEX. For the annotation purpose, we asked the annotator to annotate the table header as "Data IdString" for the event type "Deliver Data". The rest of the table rows were asked to be annotated as "Data Value".

plicity, we asked annotators to mark the headers of the table as the description of the table (Data IdString) if no better description has been specified in the email. The rest of the rows were instructed to be marked as actual data instances (Data Values). In our example, it means to mark the header "Contract Shippers Contract Date" as Data IdString and the remaining rows from "107018...Nov 1 2001" as Data Value. One could also use the row and column values to mark more complicated data instances (such as mapping each value in *Contract* column with each value in *Shippers* column and then with the values in *Contract Date* column). Modeling tables in such as way presents more informative data to the end user while complicating the task formulation by introducing a non-sequential structure. We leave this exploration to the future.

## C   Supplementary Modeling Details

### C.1   Additional Details about Sequence Labeling

**Meta Semantic Role Prediction.** As introduced in Section 2, some argument roles (e.g., the requested data attributes) have a fixed, pre-defined class space. We formulate the identification of each of such argument roles as a classification task, where the [CLS] representation will be used as in standard BERT-based classification tasks. These additional classification models will be jointly trained with the aforementioned sequence labeling model for argument extraction.

**Training and Inference.** In experiments, the trigger extraction and the argument extraction models will be trained independently. During the training time, the ground-truth trigger span and event type will be used for the argument extraction training.

At test time, given each email in an email thread, we will first apply the trigger extraction model to identify all trigger spans and their corresponding event types from the email. Then each trigger span and its type information will be fed to the argument extraction model for identifying the associated argument roles.

### C.2   Templates for End-to-End Email EE

We present the templates for the task of end-to-end email EE in Tables 8-10. All the templates begin with a sentence concerning event type with a placeholder | $trigger | for the corresponding trigger span. Following that, the templates include placeholders for the arguments specific to each event type. It is worth noting that the template contents and argument placeholders can vary depending on the meta-semantic roles involved. For instance, different templates are used when the sender expresses positive acknowledgment of an event compared to when they express negative acknowledgment. This flexibility allows for adaptable and context-aware event extraction from emails.

### C.3   Prompt Example for GPT-3.5

In Figure 8, we present the prompt design for using GPT-3.5 for event extraction. For each evaluation instance from the test set, GPT-3.5 is tasked with processing K (K=5) demonstrations, each of which consists of context, current email, and output, in addition to the task instruction and the event templates. GPT-3.5 is expected to produce a response by filling in the template for each event in the current email with its trigger and corresponding arguments.

## D   Implementation Details

### D.1   Reproducibility Details

To train sequence labeling models, we used the BERT-large-uncased with a batch size of 4 and a learning rate of 1e-5. For the generative approach, we used BART-large with a learning rate of 3e-5 and a batch size of 2. All the models were optimized using AdamW (Loshchilov and Hutter, 2017) for cross-entropy loss for 100 epochs. We tuned all the hyper-parameters on the dev set. We maintain a maximum sequence length of 512 for all our fine-tuned models. When using BART, we truncate the input sequence from the left to retain the most recent history or the recent portion of an email. During training, we implement early stopping af-

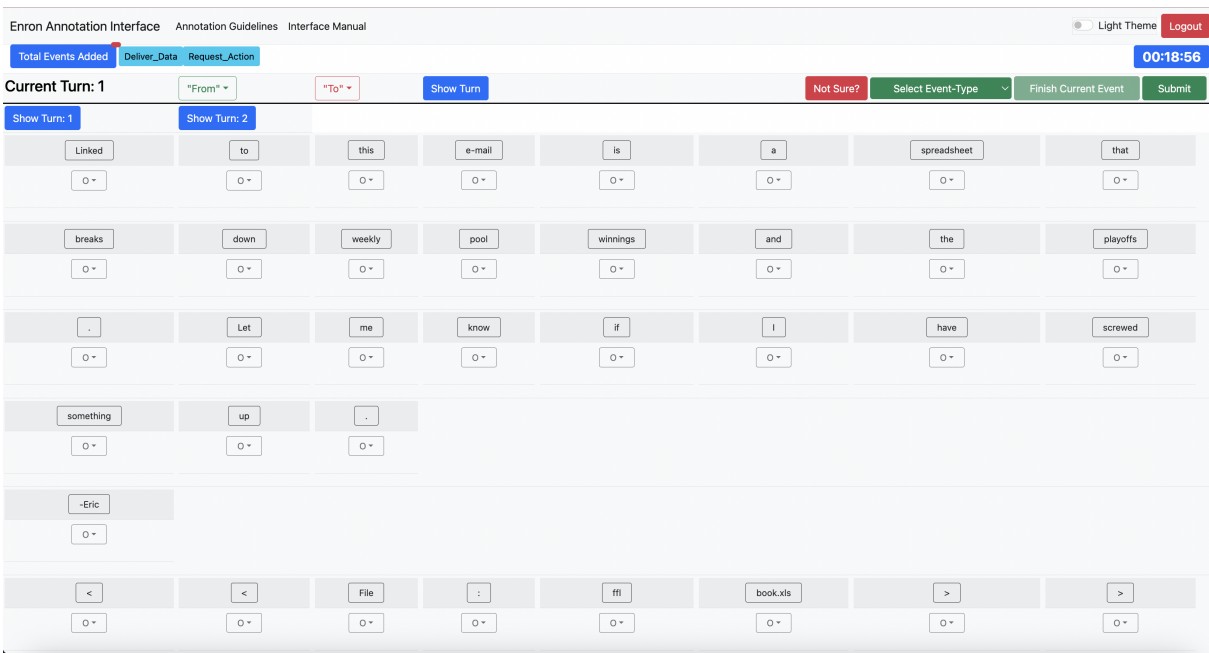

Figure 5: Our annotation interface. For each email thread, the annotators were shown each email one by one. For each email, they were tasked to select event types from a drop-down menu and directly select the event triggers by clicking on words. Next, for each word, they annotate the word with the corresponding argument role (with the default being 0 reflecting no role).

| | Example | Annotator - 1 | Annotator - 2 |
|---|---|---|---|
| **Partial Agreement** | ENA has won the bid for Lost Creek Fuel sale with a price of CIG Gas Daily plus $.03. | **Trigger:** has won 
 **Event Class:** Deliver Action Data | **Trigger:** won 
 **Event Class:** Deliver Action Data |
| | Please file the completed hardcopy in the library/fileroom. | **Trigger:** Please file 
 **Event Class:** Request Action | **Trigger:** Please file the completed hardcopy 
 **Event Class:** Request Action |
| **No Agreement** | Team , Here is an update on Oakhill : 1 . Ricki said he is sending us a T & D only contract first thing tomorrow morning. | **Trigger:** Here is an update 
 **Event Class:** Deliver Data | **Trigger:** said 
 **Event Class:** Deliver Action Data |
| | One of you UBS people with your big bonuses will have to pick it up. | **Trigger:** will have to pick it up 
 **Event Class:** Deliver Action Data | **Trigger:** pick it up 
 **Event Class:** Request Action |

Table 6: Examples for agreed and disagreed annotations. For partially agreed triggers, we keep the overlapped triggers ("*won*" and "*Please file*") while, for disagreed annotations, we remove the corresponding event and arguments annotations from the final version of the MAILEX.

ter 5 epochs, monitoring Trigger Classification for trigger extraction and Argument Classification for argument extraction.

Regarding in-context learning, we set a maximum generation length of 300 tokens with greedy decoding. All experiments were conducted using the default turbo version within the date range of 03/01/2023 to 06/13/2023. In cases where the input demonstrations exceed GPT-3.5's token limitation of 4000, we left-truncate the input sequence to ensure it fits within the specified limit. To enforce content copying and prevent the generation of extraneous information, we further adjusted the model settings. Specifically, we set the temperature pa-

rameter to 0.0, which minimizes randomness in the output, and the top_p parameter to 1, which restricts the model's choices to only the most probable tokens. These settings effectively discourage the GPT-based models from generating content that is not present in the input and encourage them to focus on copying and reproducing the input contents.

Finally, for experiments involving "ground-truth triggers" with both the BART- and In-context Learning-based approaches, we feed the templates iteratively one by one.

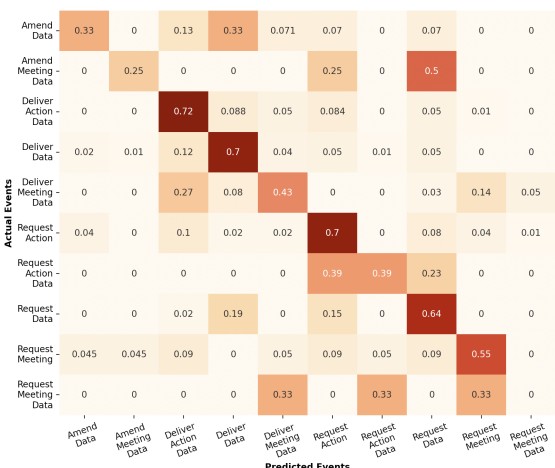

Figure 6: Confusion matrix for event type extraction using BERT-based sequence labeling. The majority of the confusion arises in noun acts, e.g., Deliver Meeting Data vs. Deliver Action Data.

### D.2 Hyperparameter Search

The BERT- and BART-based models were fine-tuned for 100 epochs using early stopping, whereby training was stopped if the validation results did not improve for 5 epochs. During the experimentation phase, we manually explored different learning rate values within the range [.01, .001, 0001, .00001, .000001] and batch sizes within the range [2, 4, 8, 16, 32]. The best model was selected based on its performance on the validation set.

### D.3 Runtime and Devices

The fine-tuned experiments were conducted on NVIDIA A100 80 GB GPU cards. Training each BERT-based model took approximately 4 hours, while the full pipeline, including training and evaluation, required approximately 8 hours. For the BART-based experiments, the training and evaluation process took approximately 12 hours. In comparison, the GPT-based experiments were completed within approximately 3-4 hours due to the time constraints imposed by the platforms used.

## E  Additional Experimental Analyses

### E.1  Classifying Noun Act Triggers

In Section 5.2.2, we discussed the models' inability to properly classify the noun acts associated with triggers. In Figure 6, we present the confusion matrix outlining the confusion between the event classes.

| | Example Email |
|---|---|
| | *Don, Attached is a detailed list of procedures and ideas on MHEB to send to the hourly crew. «MHEB_procedures.docx» «MHEB_ideas.docx»* |
| | **Ground Truth Triggers: 1)** *Attached is a detailed list* (Deliver Data) **2)** *Attached ideas* (Deliver Data) |
| BERT | **Trigger 1:** Attached is a detailed list of procedures and ideas on MHE **Trigger 2:** None |
| BART | **Trigger 1**: Attached is a detailed list **Trigger 2:** Attached ideas |
| | *Please hold Thursday , December 20th for the Board and the Committee meetings from 7:30 a.m. to 2:00 p.m. C.S.T.* |
| | **Ground Truth Triggers: 1)** *Please hold Thursday , December 20th for the Board meetings* (Request Meeting) **2)** *Please hold Thursday , December 20th for the Committee meetings* (Request Meeting) |
| BERT | **Trigger 1:** Please hold **Trigger 2:** None |
| BART | **Trigger 1:** Please hold Thursday , December 20th for the Board meetings **Trigger 2:** Please hold Thursday , December 20th for the Committee meetings |
| | *Attached we are forwarding electronic copies of the ANGTS Proposal and cover letter* |
| | **Ground Truth Triggers: 1)** *Attached we are forwarding electronic copies of cover letter* (Deliver Data) **2)** *Attached we are forwarding electronic copies of the ANGTS Proposal* (Deliver Data) |
| BERT | **Trigger 1:** Attached forwarding **Trigger 2:** None |
| BART | **Trigger 1:** Attached we are forwarding a copies of the letter **Trigger 2:** None |

Table 7: Examples of extracting shared triggers by the BERT-based sequence labeling model and the BART model. green and red indicate correct and incorrect extractions, respectively. For all the shared triggers, BERT fails to segment the trigger spans. On the contrary, BART can extract triggers when the trigger segments have a shorter distance from the conjunction "*and*" (second example). For the third example, both models fail to identify the correct trigger segments, and notably, BART adds contents that are not part of the current input. We observed that when shared triggers have a long distance in between the conjunctions (such as "*and*"), even BART struggles to retrieve them correctly.

### E.2  Shared Triggers

In Section 5.2.2, we concluded that both fine-tuned models encounter challenges in identifying shared triggers, particularly when the distance between them is significant. We show examples in Table 7. For the first two examples, where the shared triggers are relatively close, we observe that BART successfully extracts both triggers, whereas BERT either fails to segment them accurately or fails to detect them altogether. Similarly, in the last example, both the BERT and BART-based approaches struggle to identify such triggers as the distance between them increases.

### E.3  In-context Learning Analysis

**Analysis of Actual Argument Values.** We conducted an analysis of 50 randomly sampled instances where GPT-3.5 extracted erroneous arguments. We categorized these errors into the fol-

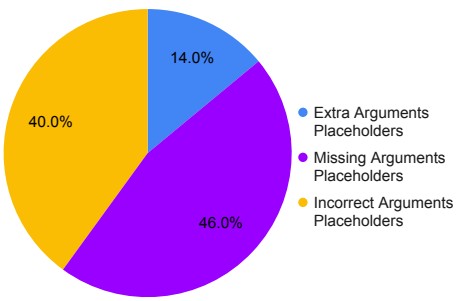

Figure 7: Error distribution while gpt-3.5-turbo generates argument placeholders.

lowing types: **1) Extra Arguments:** These errors occur when the model includes arguments that are not actual arguments for the events. For example, email signatures were mistakenly captured as member arguments for certain event types. **2) Missing Arguments:** In some cases, when generating filled-in templates with argument values, the model completely misses certain arguments and generates argument placeholders instead. **3) Semantically Correct but not Exact Match Arguments:** This type of error arises when the model attempts to summarize argument values such as Meeting Agenda or Action Description. Although semantically correct, these arguments are not recognized as exact match arguments by our evaluation script and are therefore considered incorrect. **4) Semantically Incorrect Arguments:** These involve arguments which are incorrect. While categorizing such errors we also include cases where the model adds trivial details to the arguments (such as "the" before names). **5) Semantically Incorrect and Fabricated Arguments:** Arguments that do not appear in the instructions or the current emails fall into this category and are considered both semantically incorrect and fabricated. We provide examples corresponding to each error class in Table 11.

**Analysis on Arguments Placeholders.** We also found that gpt-3.5-turbo struggles in generating consistent argument placeholders (when actual argument values are not expressed in the email). We categorized such errors into 3 categories: **1) Extra Placeholders:** While generating the templates, gpt-3.5-turbo generated more placeholders than expected. For example, for the event Deliver Action Data, it generated the template `Event Deliver Action Data is triggered by | trigger | where , | Action | is or will be performed by | Action Members | at | Time | on | Date | delivered to | Deliver Member|,`

where "Deliver Member" is an extra placeholder not provided by the event template. **2) Missing Placeholders:** Another common problem while generating the templates was identified to miss out the placeholders for arguments. For example, for the Deliver Data Events, it frequently leaves the "Data Value" placeholders as in "`Event Deliver Data is triggered by | trigger | where , | Data idString |,` (missing | Data Value |) `of | Data Type | is or will be delivered to | Deliver Members | at | Deliver Time | on | Deliver Date |`". **3) Incorrect Placeholders:** For some generated templates, we found that gpt-3.5-turbo incorrectly copies placeholders for different events than specified. For example, for the event Deliver Action Data it generated a template "`Event Deliver Action Data is triggered by | trigger | where , | Action Description | is or will be performed by | Action Members | at | Context: Action Time | on | Context: Action Date |`" which contains "Context:" label before date and time that are not part of the Deliver Action Data templates.

We randomly sampled 50 generated templates with such errors and plot an error distribution chart showing errors while generating placeholders for arguments. As Figure 7 depicts, most of the errors were made because of missing the correct or incorrect placeholders (Category 2 and 3 above).

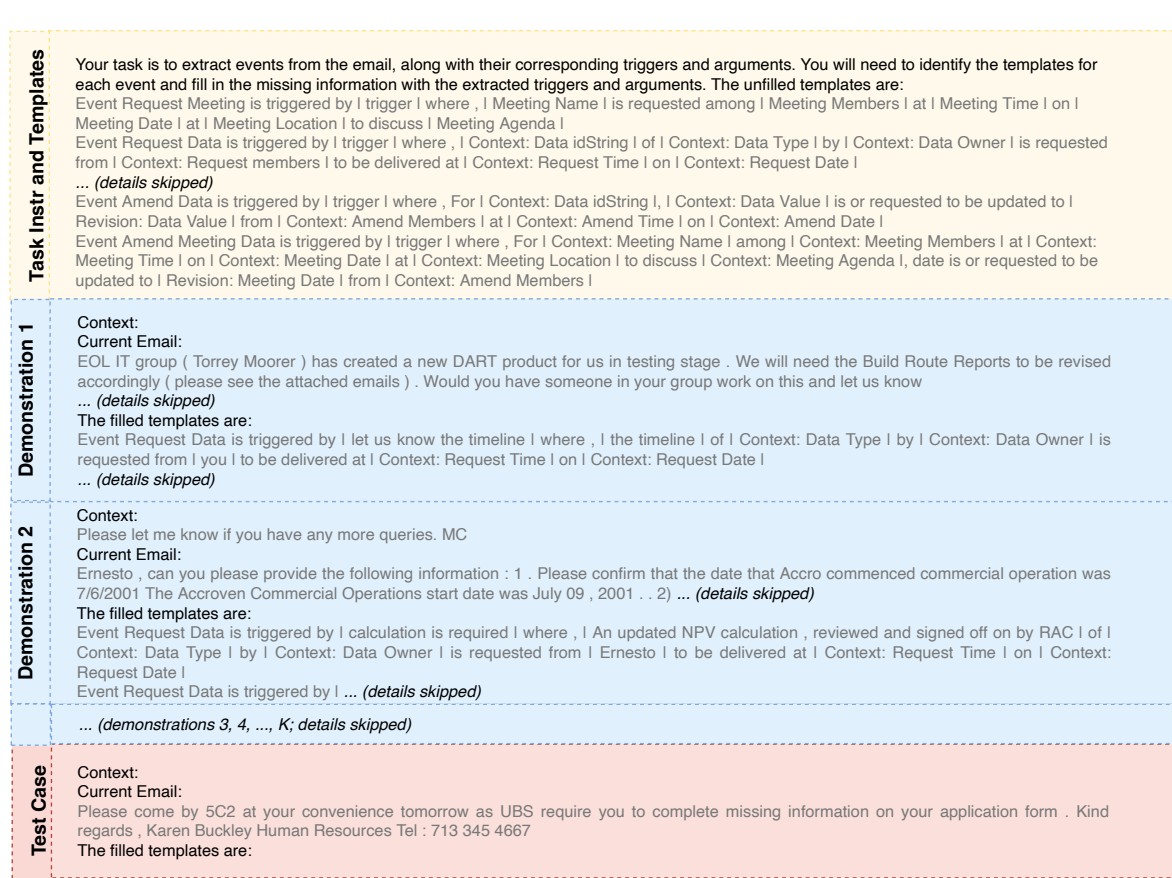

Figure 8: Prompt for event extraction using GPT-3.5. In experiments, K=5, and we ensure that the selected 5 demonstrations cover all event types and arguments.

| Event Type | Template | Example |
|---|---|---|
| Request Meeting | Event Request Meeting is triggered by \| trigger \| where, \| Meeting \| is requested among \| Meeting Members \| at \| Time \| on \| Date \| at \| Location \| to discuss \| Agenda \| | **Example:** *[Alice]*Meeting Members *has proposed a meeting on [Tuesday]*Meeting Date.

**Template:** Event Request Meeting is triggered by \| *proposed a meeting* \| where, \| *Meeting* \| is requested among \| *Alice* \| at \| Time \| on \| *Tuesday* \| at \| Location \| to discuss \| agenda \| |
| Request Data | ˜ *When ReqAttr is Value:* Event Request Data is triggered by \| trigger \| where, \| Data \| of \| Type \| by \| Owner \| is requested from \| Request Members \| to be delivered at \| Time \| on \| Date \|; 
 ˜ *When ReqAttr is Data Owner:* Event Request Data is triggered by \| trigger \| where, *Owner* of \| Data \| of \| Type \| is requested from \| Request Members \| to be delivered at \| Time \| on \| Date \| | **Example:** *Please send me [the summary of our meeting]*Data IdString (Request Attribute: Data Value);

**Template:**Event Request Data is triggered by \| *Please send me the summary* \| where, \| *the summary of our meeting* \| of \| Type \| by \| Owner \| is requested from \| Request Members \| to be delivered at \| Time \| on \| Date \| |
| Request Action | Event Request Action is triggered by \| trigger \| where, \| Action \| is requested from \| Action Members \| at \| Time \| on \| Date \| | **Example:***Please [approve Alice's travel request]*Action Description.

**Template:**Event Request Action is triggered by \| *Please approve* \| where, \| *approve Alice's travel request* \| is requested from \| Action Members \| at \| Time \| on \| Date \| |
| Request Action Data | ˜ *When ReqAttr is Action Members:* Event Request Action Data is triggered by \| trigger \| where, *Action Members* is requested for \| Action \| at \| Time \| on \| Date \| from \| Request Members \|; 
 ˜ *When ReqAttr is Action Date:* Event Request Action Data is triggered by \| trigger \| where, *Date* is requested for \| Action \| by \| Action Members \| at \| Time \| from \| Request Members \|; 
 ˜ *When ReqAttr is Action Time:* Event Request Action Data is triggered by \| trigger \| where, *Time* is requested for \| Action \| by \| Action Members \| on \| Date \| from \| Request Members \|; 
 ˜ *When ReqAttr is Action Description:* Event Request Action Data is triggered by \| trigger \| where, *Action Description* is requested for \| Action \| by \| Action Members \| on \| Date \| from \| Request Members \| | **Example:** *Who [approved the travel request]*Action Description? (Request Attribute: Action Members)

**Template:** Event Request Action Data is triggered by \| *Who approved* \| where, *Action Members* is requested for \| *approved the travel request* \| at \| Time \| on \| Date \| from \| Request Members \| |
| Request Meeting Data | ˜ *When ReqAttr is Meeting Members:* Event Request Meeting Data is triggered by \| trigger \| where, *Meeting Members* is requested for \| Meeting \| at \| Time \| on \| Date \| at \| Location \| to discuss \| Agenda \| from \| Request Members \|; 
 ˜ *When ReqAttr is Date:* Event Request Meeting Data is triggered by \| trigger \| where, *Date* is requested for \| Meeting \| among \| Meeting Members \| at \| Time \| at \| Location \| to discuss \| Agenda \| from \| Request Members \|; 
 ˜ *When ReqAttr is Time:* Event Request Meeting Data is triggered by \| trigger \| where, *Time* is requested for \| Meeting \| among \| Meeting Members \| on \| Date \| at \| Location \| to discuss \| Agenda \| from \| Request Members \|; 
 ˜ *When ReqAttr is Location:* Event Request Meeting Data is triggered by \| trigger \| where, *Location* is requested for \| Meeting \| among \| Meeting Members \| at \| Time \| on \| Tuesday \| to discuss \| Agenda \| from \| Request Members \|; 
 ˜ *When ReqAttr is Agenda:* Event Request Meeting Data is triggered by \| trigger \| where, *Agenda* is requested for \| Meeting \| among \| (Meeting) Members \| at \| Time \| on \| Date \| at \| Location \| from \| Request Members \| | **Example:** *Where is [the meeting]*Meeting Name *on [Tuesday]*Meeting Date? (Request Attribute: Meeting Location)

**Example:** Event Request Meeting Data is triggered by \| *Where is the meeting* \| where, *Location* is requested for \| *the meeting* \| among \| Meeting Members \| at \| Time \| on \| *Tuesday* \| to discuss \| Agenda \| from \| Request Members \| |

Table 8: Generation templates for end-to-end Request event extraction.

| Event Type | Template | Example |
|---|---|---|
| Deliver Data | ˜ *When Confirmation is Positive:* Event Deliver Data is triggered by \| trigger \| where, \| Data \|, \| Value \| of \| Type \| is or will be delivered to \| Deliver Members \| at \| Time \| on \| Date \|; 
 ˜ *When Confirmation is Negative:* Event Deliver Data is triggered by \| trigger \| where, \| Data \|, \| Value \| of \| Type \| is not or will not be delivered to \| Deliver Members \| at \| Time \| on \| Date \|; 
 ˜ *When Confirmation is Unsure:* Event Deliver Data is triggered by \| trigger \| where, \| Data \|, \| Value \| of \| Type \| is or will probably be delivered to \| Deliver Members \| at \| Time \| on \| Date \| | **Example:** *Attached for your review [the summary of our meeting]$_{Data\ IdString}$.* 

 **Template:** Event Deliver Data is triggered by \| *Attached the summary* \| where, \| *the summary of our meeting* \|, \| Value \| of \| Type \| is or will be delivered to \| Deliver Members \| at \| Time \| on \| Date \| |
| Deliver Action Data | ˜ *When Confirmation is Positive:* Event Deliver Action Data is triggered by \| trigger \| where, \| Action \| is or will be performed by \| Action Members \| at \| Time \| on \| Date \|; 
 ˜ *When Confirmation is Negative:* Event Deliver Action Data is triggered by \| trigger \| where, \| Action \| is not or will not be performed by \| Action Members \| at \| Time \| on \| Date \|; 
 ˜ *When Confirmation is Unsure:* Event Deliver Action Data is triggered by \| trigger \| where, \| Action \| is probably or will probably be performed by \| Action Members \| at \| Time \| on \| Date \| | **Example:** *[Alice]$_{Action\ Members}$ has agreed to [deliver mail]$_{Action\ Description}$.* (Confirmation: Positive) 

 **Template:** Event Deliver Action Data is triggered by \| *agreed* \| where, \| *deliver mail* \| is or will be performed by \| *Alice* \| at \| Time \| on \| Date \| |
| Deliver Meeting Data | ˜ *When Confirmation is Positive:* Event Deliver Meeting Data is triggered by \| trigger \| where, \| Meeting \| is or will be attended by \| Meeting Members \| at \| Time \| on \| Date \| at \| Location \| to discuss \| Agenda \|; 
 ˜ *When Confirmation is Negative:* Event Deliver Meeting Data is triggered by \| trigger \| where, \| Meeting \| is not or will not be attended by \| Meeting Members \| at \| Time \| on \| Date \| at \| Location \| to discuss \| Agenda \|; 
 ˜ *When Confirmation is Unsure:* Event Deliver Meeting Data is triggered by \| trigger \| where, \| Meeting \| is probably or will probably be attended by \| Meeting Members \| at \| Time \| on \| Date \| at \| Location \| to discuss \| Agenda \| | **Example:** *[Alice]$_{Members}$ will attend the [Tuesday]$_{Date}$ [Board meeting]$_{Meeting\ Name}$.* (Confirmation: Positive) 

 **Template:** Event Deliver Meeting Data is triggered by \| *will attend meeting* \| where, \| *Board Meeting* \| is or will be attended by \| *Alice* \| at \| Time \| on \| *Tuesday* \| at \| Location \| to discuss \| Agenda \| |

Table 9: Generation templates for end-to-end Deliver event extraction.

| Event Type | Template | Example |
|---|---|---|
| Amend Data | ˜ *When Amend Action is Update:* Event Amend Data is triggered by \| trigger \| where, for \| Cnt: Data \|, \| Cnt: Value \| is or requested to be updated to \| Rev: Value \| from \| Amend Members \| at \| Time \| on \| Date \|; 
 ˜ *When Amend Action is Add:* Event Amend Data is triggered by \| trigger \| where, for \| Cnt: Data \|, \| Rev: Value \| is or requested to be added from \| Amend Members \| at \| Time \| on \| Date \|; 
 ˜ *When Amend Action is Delete:* Event Amend Data is triggered by \| trigger \| where, for \| Cnt: Data \|, \| Con: Value \| is or requested to be removed from \| Amend Members \| at \| Time \| on \| Date \| | **Example:** *Can [you]$_{Members}$ change the [budget]$_{CNT:Data\ IdString}$ from [2K]$_{CNT:Data\ Value}$ to [3K]$_{REV:Data\ Value}$* (Amend Type: Update) 

 **Template:** Event Amend Data is triggered by \| *change the budget* \| where, for \| *budget* \|, \| *2K* \| is or requested to be updated to \| *3K* \| from \| *you* \| at \| Time \| on \| Date \| |
| Amend Meeting Data | ˜ *To update meeting members:* Event Amend Meeting Data is triggered by \| trigger \| where, for \| Meeting \| among \| Cnt: Meeting Members \| at \| Cnt: Time \| on \| Cnt: Date \| at \| Cnt: Location \| to discuss \| Cnt: Agenda \|, meeting members is or requested to be updated to \| Rev: Meeting Members \| from \| Amend Members \|; 
 ˜ *To update meeting date:* Event Amend Meeting Data is triggered by \| trigger \| where, for meeting \| Meeting \| among \| Cnt: Meeting Members \| at \| Cnt: Time \| on \| Cnt: Date \| at \| Cnt: Location \| to discuss \| Cnt: Agenda \|, date is or requested to be updated to \| Rev: Date \| from \| Amend Members \|; 
 ˜ *To update meeting time:* Event Amend Meeting Data is triggered by \| trigger \| where, for meeting \| Meeting \| among \| Cnt: Meeting Members \| at \| Cnt: Time \| on \| Cnt: Date \| at \| Cnt: Location \| to discuss \| Cnt: Agenda \|, time is or requested to be updated to \| Rev: Time \| from \| Amend Members \|; 
 ˜ *To update meeting location:* Event Amend Meeting Data is triggered by \| trigger \| where, for meeting \| Meeting \| among \| Cnt: Meeting Members \| at \| Cnt: Time \| on \| Cnt: Date \| at \| Cnt: Location \| to discuss \| Cnt: Agenda \|, location is or requested to be updated to \| Rev: Location \| from \| Amend Members \|; 
 ˜ *To update meeting agenda:* Event Amend Meeting Data is triggered by \| trigger \| where, for meeting \| Meeting \| among \| Cnt: Meeting Members \| at \| Cnt: Time \| on \| Cnt: Date \| at \| Cnt: Location \| to discuss \| Cnt: Agenda \|, agenda is or requested to be updated to \| Rev: Agenda \| from \| Amend Members \|; 

 ˜ *To add meeting members:* Event Amend Meeting Data is triggered by \| trigger \| where, for \| Meeting \| among \| Cnt: Meeting Members \| at \| Cnt: Time \| on \| Cnt: Date \| at \| Cnt: Location \| to discuss \| Cnt: Agenda \|, meeting members \| Rev: Meeting Members \| is or requested to be added from \| Amend Members \|; 
 ˜ *To add meeting date:* Event Amend Meeting Data is triggered by \| trigger \| where, for \| Meeting \| among \| Cnt: Meeting Members \| at \| Cnt: Time \| at \| Cnt: Location \| to discuss \| Cnt: Agenda \|, date \| Rev: Date \| is or requested to be added from \| Amend Members \|; 
 ˜ *To add meeting time:* Event Amend Meeting Data is triggered by \| trigger \| where, for \| Meeting \| among \| Cnt: Meeting Members \| on \| Cnt: Date \| at \| Cnt: Location \| to discuss \| Cnt: Agenda \|, time \| Rev: Time \| is or requested to be added from \| Amend Members \|; 
 ˜ *To add meeting location:* Event Amend Meeting Data is triggered by \| trigger \| where, for \| Meeting \| among \| Cnt: Meeting Members \| at \| Cnt: Time \| on \| Cnt: Date \| to discuss \| Cnt: Agenda \|, location \| Rev: Location \| is or requested to be added from \| Amend Members \|; 
 ˜ *To add meeting agenda:* Event Amend Meeting Data is triggered by \| trigger \| where, for \| Meeting \| among \| Cnt: Meeting Members \| at \| Cnt: Time \| on \| Cnt: Date \| at \| Cnt: Location \|, agenda \| Rev: Agenda \| is or requested to be added from \| Amend Members \|; 

 ˜ *To remove meeting members:* Event Amend Meeting Data is triggered by \| trigger \| where, for \| Meeting \| among \| Cnt: Meeting Members \| at \| Cnt: Time \| on \| Cnt: Date \| at \| Cnt: Location \| to discuss \| Cnt: Agenda \|, meeting members \| Rev: Meeting Members \| is or requested to be removed from \| Amend Members \| | **Example:** *Can we reschedule [the meeting]$_{CNT:Meeting\ Name}$ on [Tuesday]$_{CNT:Meeting\ Date}$ to [Friday]$_{REV:Meeting\ Date}$ instead?* (Amend Type: Update) 

 **Template:** Event Amend Meeting Data is triggered by \| *reschedule the meeting* \| where, for meeting \| *the meeting* \| among \| Cnt: Meeting Members \| at \| Cnt: Time \| on \| *Cnt: Tuesday* \| at \| Cnt: Location \| to discuss \| Cnt: Agenda \|, date is or requested to be updated to \| *Friday* \| from \| Amend Members \| |

Table 10: Generation templates for end-to-end Amend event extraction.

| Error Category | |
|---|---|
| | **Example:** *Laura Giambrone discovered a PNR balance of 10,000 Dth on the above noted contract* |
| | **Gold Template:** Event Deliver Action Data is triggered by \| discovered \| where , \| discovered a PNR balance of 10,000 Dth on the above noted contract \| is or will be performed by \| Laura Giambrone \| at \| Action Time \| on \| Action Date \| |
| **Semantically Incorrect Arguments** | **GPT-3.5-turbo:** Event Deliver Action Data is triggered by \| discovered \| where , \| `Laura Giambrone discovered a PNR balance of 10,000 Dth on the above noted contract` \| is or will be performed by \| Laura Giambrone \| at \| Action Time \| on \| Action Date \| |
| | **GPT-3.5 davinci-003:** Event Deliver Action Data is triggered by \| discovered \| where , \| discovered a PNR balance of 10,000 Dth \| is or will be performed by \| Laura Giambrone \| at \| Action Time \| on \| Action Date \| |
| | **Example:** *How about we all go to dinner at Mark 's on Sunday night ? How does that sound ? -Eric* |
| | **Gold Template:** Event Request Meeting is triggered by \| How about we all go to dinner \| where , \| Meeting Name \| is requested among \| we \| at \| night \| on \| Sunday \| at \| Mark 's \| to discuss \| dinner \| |
| **Semantically Incorrect and Fabricated Arguments** | **GPT-3.5-turbo:** Event Request Meeting is triggered by \| How about we all go to dinner \| where , \| `there is no trigger or context in the email to fill in any of the event templates.` |
| | **GPT-3.5 davinci-003:** Event Request Meeting is triggered by \| How about we all go to dinner \| where , \| Meeting Name \| is requested among \| we all \| at \| Meeting Time \| on \| Sunday night \| at \| Meeting Location \| to discuss \| How does that sound ? \| |
| | **Example:** *Hi Shelley I have a meeting at 2:00-3:30 and am meeting Kevin at 4:00 . Can I come to your office at 3:30 ?* |
| | **Gold Template:** Event Amend Meeting Data is triggered by \| Can I come to your office at 3:30 ? \| where , For \| Context: Meeting Name \| among \| I \| at \| Context: Meeting Time \| on \| Context: Meeting Date \| at \| your office \| to discuss \| Context: Meeting Agenda \|, time is or requested to be updated to \| 3:30 \| from \| Context: Amend Members \| |
| **Semantically Correct but not Exact Matched Arguments** | **GPT-3.5-turbo:** Event Amend Meeting Data is triggered by \| Can I come to your office at 3:30 ? \| where , For \| Meeting Name \| among \| Meeting Members \| at \| Context: Meeting Time \| on \| Context: Meeting Date \| at \| `Shelley's office` \|, time is or requested to be updated to \| 3:30 \| from \| Context: Amend Members \| |
| | **GPT-3.5 davinci-003:** Event Amend Meeting Data is triggered by \| Can I come to your office at 3:30 ? \| where , For \| the meeting \| among \| I and Kevin \| at \| Context: Meeting Time \| on \| Context: Meeting Date \| at \| Context: Meeting Location \|, agenda \| discuss \| is or requested to be added from \| Shelley \| |
| | **Example:** *Where is the website for ERCOT ? JD* |
| | **Gold Template:** Event Request Data is triggered by \| Where is the website \| where , \| the website for ERCOT \| of \| Context: Data Type \| by \| Context: Data Owner \| is requested from \| Context: Request members \| to be delivered at \| Context: Request Time \| on \| Context: Request Date \| |
| **Extra Arguments** | **GPT-3.5-turbo:** Event Request Data is triggered by \| Where is the website \| where , \| the website for ERCOT \| of \| Context: Data Type \| by \| `JD` \| is requested from \| Rika \| to be delivered at \| Context: Request Time \| on \| Context: Request Date \| |
| | **GPT-3.5 davinci-003:** Event Request Data is triggered by \| Where is the website \| where , \| website for ERCOT \| of \| Context: Data Type \| by \| Context: Data Owner \| is requested from \| `JD` \| to be delivered at \| Context: Request Time \| on \| Context: Request Date \| |
| | **Example:** *Attached is a 2 nd draft of the qtrly letter reflecting more of my changes , ... < < AAP memo for qtrly repts- ( rev2-rj ) .doc > >* |
| | **Gold Template:** Event Deliver Data is triggered by \| Attached is a 2 nd draft \| where , \| 2 nd draft of the qtrly letter reflecting more of my changes \|, \| AAP memo for qtrly repts- ( rev2-rj ) .doc \| of \| Data Type \| is or will be delivered to \| Deliver Members \| at \| Deliver Time \| on \| Deliver Date \| |
| **Missing Arguments** | **GPT-3.5-turbo:** Event Deliver Data is triggered by \| Attached is a 2 nd draft \| where , \| 2nd draft \|, \| `Data Value` \| of \| Data Type \| is or will be delivered to \| Deliver Members \| at \| Deliver Time \| on \| Deliver Date \| |
| | **GPT-3.5 davinci-003:** Event Deliver Data is triggered by \| Attached is a 2 nd draft \| where , \| 2 nd draft of the qtrly letter \|, \| Data Value \| of \| Data Type \| is or will be delivered to \| Deliver Members \| at \| Deliver Time \| on \| Deliver Date \| |

Table 11: Analysis of in-context learning-based approaches when ground truth triggers are fed to variants of GPT-3.5. The errors made by models are highlighted in red and the ground-truth templates are highlighted in gold.