# OpenReview forum: "MailEx: Email Event and Argument Extraction"
_EMNLP/2023/Conference — EMNLP 2023 Main_

### Official Review · Reviewer_UBTU · 2023-08-02

**Soundness:** 4

**Excitement:**

4: Strong: This paper deepens the understanding of some phenomenon or lowers the barriers to an existing research direction.

**Paper Topic And Main Contributions:**

This paper studies the event extraction task in email threads. Event extraction comprises of event trigger identification, event trigger classification, argument identification, and argument classification. The authors define a taxonomy of event types (verb + noun acts) and argument types. They apply this taxonomy to annotate 1.5K email threads between Enron employees from the Enron email thread dataset. The experiment sequence labeling, sequence to sequence, and few-shot generative approaches on the event extraction tasks. The authors observe that sequence labeling models perform the best on this closed domain task. Finding non-contiguous triggers and non-named entity arguments are challenging.

**Questions For The Authors:**

Q1. How did you decide upon the three verb acts: request, deliver, and amend? Were their other email actions which were ignored?
Q2. Do you think pretraining the language modeling objective of the models on conversation data would help your cause? If so, why did you not use it? I am assuming such specialized pretrained models are readily available.

**Reasons To Accept:**

This paper makes several contributions towards the information extraction field. First, they define a taxonomy for event extraction in corporate email threads. Second, they release the MailEx dataset which would be a valuable resource for evaluating IE models. Third, they contribute novel event extraction models and highlight the deficiencies of few-shot LLMs.

**Reasons To Reject:**

I did not find any major technical weaknesses in the paper. However, the authors could improve the language of certain sections such as the end-to-end extraction model discussion to improve comprehension. I would also recommend that the authors show statistical test results to support their model comparison claims.

**Reproducibility:**

4: Could mostly reproduce the results, but there may be some variation because of sample variance or minor variations in their interpretation of the protocol or method.

**Reviewer Confidence:**

4: Quite sure. I tried to check the important points carefully. It's unlikely, though conceivable, that I missed something that should affect my ratings.

---

> ### Author Rebuttal · Authors · 2023-08-29
>
> We extend our gratitude for your valuable time and thoughtful assessment. Your recognition of the utility of MailEx, along with its distinctive complexities in executing event extraction from emails, is deeply acknowledged. Furthermore, we express our gratitude for your engagement with our comprehensive study using novel baselines and highlighting the shortcomings of LLMs in a few-shot setting.
>
> **1. Suggestions on improving the writing of sections and statistical tests to further support model comparison.**
>
> Thank you for the suggestion. We will refine the writing of section 4.3 to improve comprehension.
>
> For statistical tests, we have performed a Wilcoxon signed-rank test (Wilcoxon, 1945), which is used when comparing two matched samples. We compared BERT’s and BART’s trigger classification and argument classification F1 performance on the dev set, setting a significance level $\alpha$ = 0.05.
>
> We observed for triggers, with a p-value of 0.0329, that BERT is significantly better than BART. However, for arguments, with a p-value of 0.287, we conclude that there is no statistically significant difference between BART and BERT. These results support our model comparison claims made in lines 417-422. We will add the results to our revised draft.
>
> **2. How did you identify the three verb acts: request, deliver, and amend? Did you ignore other email actions?**
>
> The verb acts, i.e., request, deliver, and amend, are inspired by the previous work on email speech act (Cohen et al., 2004). In the original paper, Cohen et al. proposed a total of 9 verb acts but then reduced them to a subset of 5 verbs based on their frequency. We further trim them off by (1) merging “propose” with “request” into “request” due to their shared purpose of encouraging recipients to undertake actions, whether through suggested courses or direct appeals for assistance, and (2) merging “commit” and “deliver” into “deliver” but introducing a “confirmation” slot (lines 147-155) to indicate the intent of commitment.
>
>
> **3.  Pretraining the language modeling objective of the models on conversation data**
>
>
>
> This is a good suggestion, but it seems to be a very open-ended question.
>
> First of all, we identify a domain gap between the conversation data that existing chat LMs such as Vicuna (Chiang, Wei-Lin, et al.), FastChat (Zheng et al.), and Koala (Geng et al.) have been pre-trained on, and the target email conversation. That is, the business-dominant context and the formal language usage in EmailEx could be very different from the conversation data used to pre-train these chat LMs.
>
> Alternatively, instead of using these pre-trained chat LMs, one could also pre-train the base LM (i.e., BERT or BART in our work) on conversational email data, before fine-tuning them in the annotated EE task. We think that this could help improve the model's performance and would be excited to try it in our future work!
>
> Finally, we note that in our in-context learning experiment, gpt-3.5-turbo is better optimized for chats compared with text-davinci-003, but the former performs worse than the latter. This shows the challenge of Email EE beyond simply conversational language understanding, e.g., copying exact long-text argument spans without hallucination.
>
> In this work, we focused on introducing the dataset and analyzing the performance of classic EE models on the new task and hence were not able to experiment with the suggested idea thoroughly, but we are excited to investigate this research question carefully in our future work.
>
> **References:**
>
> Cohen, William, Vitor Carvalho, and Tom Mitchell. "Learning to classify email into “speech acts”." Proceedings of the 2004 Conference on Empirical Methods in Natural Language Processing. 2004.
>
> Chiang, Wei-Lin, et al. "Vicuna: An open-source chatbot impressing gpt-4 with 90%* chatgpt quality." See https://vicuna. lmsys. org (accessed 14 April 2023) (2023).
>
> Lianmin Zheng, Wei-Lin Chiang, Ying Sheng, Siyuan Zhuang, Zhanghao Wu, Yonghao Zhuang, Zi Lin, Zhuohan Li, Dacheng Li, Eric. P Xing, Hao Zhang, Joseph E. Gonzalez, & Ion Stoica. (2023). Judging LLM-as-a-judge with MT-Bench and Chatbot Arena.
>
>
>
> Xinyang Geng, Arnav Gudibande, Hao Liu, Eric Wallace, Pieter Abbeel, Sergey Levine, & Dawn Song. (2023). Koala: A Dialogue Model for Academic Research.
>
>
>
> Frank Wilcoxon. 1945. Individual comparisons by ranking methods. Biometrics bulletin 1(6):80–83.

---

### Official Review · Reviewer_LSSd · 2023-08-07

**Soundness:** 4

**Excitement:**

3: Ambivalent: It has merits (e.g., it reports state-of-the-art results, the idea is nice), but there are key weaknesses (e.g., it describes incremental work), and it can significantly benefit from another round of revision. However, I won't object to accepting it if my co-reviewers champion it.

**Paper Topic And Main Contributions:**

This paper focuses on the task Event and Argument Extraction in Email domain. This paper designs a new dataset that annotates event types and arguments on emails, and they evaluate three different baselines on the dataset: BERT-based sequence labeling, BART-based e2e extraction, and GPT3.5, and they show that these methods' performance is low.

**Questions For The Authors:**

How many domains do these emails come from? All business emails from Enron?

Have you tried to annotate more fine-grained event types? If yes, what makes you decide not to include them?

**Reasons To Accept:**

A new dataset for Email domain itself is interesting and could be important for real applications.
The proposed baseline methods do not perform very well on this dataset, which leaves a large space for future works to further improve.

**Reasons To Reject:**

Though the dataset is very interesting, the granularity of proposed event type is not fine enough. All the event type is about a specific task for business, like "schedule a meeting" and "request data". This is more like emails in a company. However, if we are talking about event in emails, emails should also cover all the event types of other texts. People can talk about any events in email, but the key difference is: the context of email is more daily, which may make the task harder. In my opinion, the authors should consider extend the type set to include all the event types that other datasets use.

The result of GPT-3.5 could be problematic, since the event template provided in the prompt is not easy to understand.

**Reproducibility:**

4: Could mostly reproduce the results, but there may be some variation because of sample variance or minor variations in their interpretation of the protocol or method.

**Reviewer Confidence:**

4: Quite sure. I tried to check the important points carefully. It's unlikely, though conceivable, that I missed something that should affect my ratings.

---

> ### Author Rebuttal · Authors · 2023-08-29
>
> Thank you for your time and thoughtful comments on our work!
>
>
> **1.  Comment on the proposed event type being not fine enough and based on business context**
>
> Our taxonomy is inspired by Cohen et. al. (2004), which is fine-grained for email communication in a task-oriented setting and is not limited to only business. We emphasize in lines 100-115 that the focus of this work is to extract commonly seen events from daily email communications. Although business communication is intuitively a major use case of emails, as elaborated in our response to Q3, Enron itself contains a significant portion of emails sent for personal purposes. Based on our preliminary study on 3 email corpora, we keep our focus on the 10 most common event types enabled by three verb and three noun acts. Some of our event types are very broadly defined; for example, a “Meeting” in our definition can be both a formal meeting (business or not, in-person or via phone calls) and an informal social gathering (e.g., birthday party, a camping trip, impromptu outing, etc.).
>
> While most context of MailEx is about business usage, we highlight that (1) as stated above, our event schema itself is not limited to the business context, and that (2) to the best of our knowledge, Enron is the only open-source, publicly available, and large-scale email corpus. Despite the context limitation (which we also discussed in Limitations), we believe MailEx is still a valuable resource for future research in conversational, email EE.
>
> Finally, while creating a holistic event ontology by “includ(ing) all the event types that other datasets use” could be an interesting idea, this is not trivial as it requires schema-matching across different datasets and their contexts. Training EE models to understand such a comprehensive event ontology will also require a sufficiently complex dataset, which as far as we know is not available. Instead, our focus in this work is to propose a new task of “conversational EE” in the email context, providing baseline models and inspiring future research to look into this rarely explored field.
>
> **2. Concern about event templates being not easy to understand and their impact on GPT-3.5’s result.**
>
> The templates designed for performing EE on emails have been carefully designed, inspired by previous research such as Du et. al (2022). These templates are easy to understand and would not be the reason for GPT-3.5’s low performance. As evidenced by Table 2, the performance of BART-based approaches using the same templates is on par with the sequence labeling-based approaches.
>
> Instead, we hypothesize the low performance of the GPT-based model could have occurred due to the intrinsic inability of LLMs to understand the nuanced languages and their implied event information. In Section 5.2.5, we provided more careful categorization and analysis of the errors made by GPT-3.5.
>
>
>
> **3. How many domains do these emails come from? All business emails from Enron?**
>
> Most of the emails in MailEx are in a business context, although sometimes the emails could be personal as well (e.g., exchanging thoughts about their job or work with their relatives). In fact, prior work (Alkhereyf and Rambow, 2017) found that a significant portion of Enron emails were written for personal purposes.
>
>
> **4. Annotation and inclusion of more fine-grained event types.**
>
> We first clarify that our current event schema definition is already fine-grained. For example, we distinguish among Meeting Data, Action Data, and other general types of Data, as Reviewer uHNK also noticed. Our schema follows the email speech acts of Cohen et al. (2004) and retains only the most frequent event types in the Enron dataset. There are also noun acts such as “Opinion” and “Thanking” which we drop from our schema because they do not directly connect to “events”.
>
> Including additional event types that are rare in Enron is not very meaningful due to the lack of data context. While this is a limitation caused by the choice of Enron, we note again the limited email data resources available for NLP research, and that MailEx is the first dataset utilizing available resources toward filling the gap of conversational, email event extraction research.
>
> **References:**
>
> Cohen, William, Vitor Carvalho, and Tom Mitchell. "Learning to classify email into “speech acts”." Proceedings of the 2004 Conference on Empirical Methods in Natural Language Processing. 2004.
>
> Du, X., Li, S., & Ji, H. (2022). Dynamic Global Memory for Document-level Argument Extraction. In Proceedings of the 60th Annual Meeting of the Association for Computational Linguistics (Volume 1: Long Papers) (pp. 5264–5275). Association for Computational Linguistics
>
> Alkhereyf, Sakhar, and Owen Rambow. "Work hard, play hard: Email classification on the Avocado and Enron corpora." Proceedings of TextGraphs-11: the Workshop on Graph-based Methods for Natural Language Processing. 2017.

---

### Official Review · Reviewer_FtbS · 2023-08-08

**Soundness:** 4

**Excitement:**

4: Strong: This paper deepens the understanding of some phenomenon or lowers the barriers to an existing research direction.

**Paper Topic And Main Contributions:**

The paper presents an email event extraction dataset that facilitates comprehensive downstream tasks on email data. The task is novel and has not been addressed previously in the literature. The authors first propose a taxonomy of the event extraction dataset before annotating the dataset. To understand the challenges of the dataset, the authors have proposed three types of experiments and reported the results. The authors also discuss some of the unique challenges of email event extraction.

**Reasons To Accept:**

1. The dataset is novel and has addressed some of the critical characteristics of email data. The proposed dataset can be a valuable study resource in this unexplored research area.
2. The authors have adequately described all the sections of the manuscript, including the taxonomy of the dataset, annotation details, and a detailed analysis of the dataset statistics and unique characteristics.
3. The authors have described three types of baselines for the proposed dataset and compared the results. The authors have provided the experimental setting of the models that improve the reproducibility of the baseline models.
4. The authors also performed the qualitative analysis of the results based on the unique challenges of event and argument extraction of the proposed dataset.


**Reasons To Reject:**

1. No reason to reject the work.

**Reproducibility:**

4: Could mostly reproduce the results, but there may be some variation because of sample variance or minor variations in their interpretation of the protocol or method.

**Reviewer Confidence:**

4: Quite sure. I tried to check the important points carefully. It's unlikely, though conceivable, that I missed something that should affect my ratings.

---

> ### Author Rebuttal · Authors · 2023-08-29
>
> Thank you for your review! We deeply appreciate your recognition of the usability of the MailEx dataset and our analyses of unique traits and challenges in performing email event extraction.

---

### Official Review · Reviewer_uHNK · 2023-08-20

**Soundness:** 3

**Excitement:**

3: Ambivalent: It has merits (e.g., it reports state-of-the-art results, the idea is nice), but there are key weaknesses (e.g., it describes incremental work), and it can significantly benefit from another round of revision. However, I won't object to accepting it if my co-reviewers champion it.

**Paper Topic And Main Contributions:**

The paper presents the first dataset for event extraction in the email domain, with a new taxonomy covering 10 event types and 76 arguments. The authors implements three models, including sequence labeling, template-based generative extraction, and few-shot in-context learning, on the dataset and make comprehensive analysis on the challenges of the task.

**Questions For The Authors:**

1. How do you deal with the partially agreed annotations? What's the criteria for a better annotation?
2. For the case in line 425, have you tried to limit the vocabulary space to the words in the email context?
3. In line 404-408, do you consider two spans are partial matching if they overlap with each other?
4. In the BART model, does the input context contain the concatenation of templates for all events? In Li et al., 2021, they firstly identify the event triggers and only include the template of a specific event in the input context, extracting arguments for one event at a time.

**Reasons To Accept:**

The paper has several strengths:
- Overall, it is well-written with a clear and well-motivated introduction.
- The high-quality annotated dataset will facilitate event extraction research in the email domain and prompt the study of email assistance.

**Reasons To Reject:**

1. There is a lack of clarity regarding the criteria of designing the schema. For example, why differentiate the event type "Request Meeting Data" from "Request Data" as the previous event only cover 0.71% of all events?
2. The average number of words of arguments is 7.41 and some of them are non-continuous spans. This will cause difficulties and ambiguities in identifying the arguments.

**Reproducibility:**

4: Could mostly reproduce the results, but there may be some variation because of sample variance or minor variations in their interpretation of the protocol or method.

**Reviewer Confidence:**

4: Quite sure. I tried to check the important points carefully. It's unlikely, though conceivable, that I missed something that should affect my ratings.

---

> ### Author Rebuttal · Authors · 2023-08-29
>
> We first thank you for taking the time to review our paper and giving thoughtful comments!
>
> **1. Clarity regarding the criteria for designing the schema.**
>
> Our criteria for designing the event schema are inspired by Cohen et al. (2004), who defined task-oriented speech acts for email communication. We extended the definition of these speech acts to perform the task of event extraction on emails. Please refer to Section 2.1 for clarification and Appendix A.1 for definitions of the verb and noun acts used in our dataset.
>
> We differentiate the event type “Request Meeting Data” from “Request Data” to accommodate for the intrinsically different types of “Data” in email events. Specifically, our schema considers three types of data: (a) “Meeting Data”, which refers to facts related to specific meetings (e.g., meeting date, location), (b) “Action Data”, which refers to facts related to a specific action or an activity (e.g., a deadline for approving a budget request, the person who approved the request, etc.), and (c) “Data” refers to all other information irrelevant to meetings and actions, such as PDF files sent in emails.
>
> Notably, doing this allows us to easily connect the event extraction task with downstream applications; for example, when an EE model extracts meeting information, a downstream email reminder can be automatically set up to provide additional assistance. However, this will not be feasible if we simply merge all types of data information into one coarse category. While in our current dataset, only a small portion of “Request Meeting Data” events were annotated, we included this fine-grained event type to facilitate follow-up research and broader downstream applications in the future.
>
> In our revised draft, we will clarify the definition of different data types and emphasize this motivation.
>
> **2. Concern about long and non-continuous arguments.**
>
> We first clarify that, unlike triggers, arguments are always continuous spans in our dataset. As mentioned in lines 261-270, arguments can span across multiple words or sentences due to the presence of non-named entities such as “Meeting Agenda” or “Meeting Location”, but they are always continuous.
>
> Annotating long-span arguments, as the reviewer pointed out, is indeed challenging. Therefore, we provided very careful instructions to our annotators, including instructing them to annotate only a minimal span for each argument. In addition, when two annotators partially agreed on an annotated argument span, we retained their overlap, which further reduced any redundant words in each argument. Due to the space limit, we have included most of our annotation details in Appendix B, including examples of partially agreed annotations in B.2, from which we can confirm no ambiguity in the actual argument annotation, despite its long length.
>
> **3. Details on partially agreed annotations and criteria for better annotation.**
>
> For partially agreed triggers (but with agreement on the event type), we retained the overlapped word spans. For partially agreed arguments (but similarly with agreement on the event type and having overlapped trigger spans), we similarly retained the overlapped word span. When two annotators did not agree on the event type or made no overlap in their annotated triggers, we abandoned the annotations completely. We will clarify it in our revised draft.
>
> In terms of the criteria for better annotations, we included the annotation guidelines in Appendix B.1 due to space limit. In summary, to facilitate the annotation, we provided detailed definitions and examples of each event type, an intuitive interface for annotation (Figure 4), and instructions to reduce confusion. Annotators also received training before starting the formal annotation.
>
> **4. Have you tried to limit the vocabulary space to the words in the email context?**
>
> No, we have not tried the suggested solution. First, we’d like to note that limiting the decoder vocabulary to only words from the current email is not a feasible solution because the decoder has to copy words from the template as well. Second, even if we constrain the decoder to select words from both the current email and the templates, it won’t solve the low recall issue. In fact, even without any constraint (as in our current experiments), BART has learned to decode words only from these two sources (i.e., the current email and the templates), so adding the suggested constraint will not really change the model performance. In our observation, the high precision and low recall of BART in argument extraction typically happened when BART copied correct but incomplete argument spans, and those spans are typically long-length, non-named entity spans. To solve the issue, future work should look into how to improve BART in understanding non-named entity arguments.
>
> **5. In L404-408, do you consider two spans a partial match if they overlap with each other?**
>
> Yes, we consider overlapped spans as partially matched. As mentioned in lines 404-408, we follow this strategy to encourage a more fair comparison (such as both the spans “The president” and “president” will be acceptable). When there is only a partial match (as opposed to a complete, exact match), the model will incur a lower argument identification/classification performance. Similar strategies have been adopted in previous studies such as Li et al. (2021).
>
> **6. For BART, do you input all the event templates?**
>
> Yes, for BART, the input context contains the concatenation of templates for all events. We designed it for the interest of end-to-end EE modeling. We hypothesized that triggers and arguments should correlate with each other. Extracting them in an end-to-end fashion may improve the overall performance as has been demonstrated in previous works (Hsu et al., 2022, Lu et al., 2021, and Du et al., 2022., inter alia).
>
> In contrast, Li et al. (2021) follow a pipeline approach similar to our BERT-based method, except that for argument extraction they adopted BART rather than BERT. While this is also feasible, it could lead to cascading of errors (Du et. al, 2021) as errors from the triggers extraction stage could not be used to reinforce the argument learning.
>
> Notably in Table 2, we did provide results comparing BERT and BART for argument extraction, when both were fed with ground-truth triggers. In this experiment, we only input the targeted event template and gold trigger to BART and expect it to extract arguments for one event at a time. It is shown that BART obtained a worse argument classification F1 than BERT.
>
>
> **References:**
>
> Hsu*, I.H., Huang*, K.H., Boschee, E., Miller, S., Natarajan, P., Chang, K.W., & Peng, N. (2022). DEGREE: A Data-Efficient Generative Event Extraction Model. In Proceedings of the 2022 Conference of the North American Chapter of the Association for Computational Linguistics: Human Language Technologies (NAACL).
>
> Lu, Y., Lin, H., Xu, J., Han, X., Tang, J., Li, A., Sun, L., Liao, M., & Chen, S. (2021). Text2Event: Controllable Sequence-to-Structure Generation for End-to-end Event Extraction. In Proceedings of the 59th Annual Meeting of the Association for Computational Linguistics and the 11th International Joint Conference on Natural Language Processing (Volume 1: Long Papers) (pp. 2795–2806). Association for Computational Linguistics.
>
> Du, X., Li, S., & Ji, H. (2022). Dynamic Global Memory for Document-level Argument Extraction. In Proceedings of the 60th Annual Meeting of the Association for Computational Linguistics (Volume 1: Long Papers) (pp. 5264–5275). Association for Computational Linguistics
>
> Du, X., Rush, A., & Cardie, C. (2021). Template Filling with Generative Transformers. In Proceedings of the 2021 Conference of the North American Chapter of the Association for Computational Linguistics: Human Language Technologies (pp. 909–914). Association for Computational Linguistics.
>
> Li, Sha, Heng Ji, and Jiawei Han. "Document-Level Event Argument Extraction by Conditional Generation." Proceedings of the 2021 Conference of the North American Chapter of the Association for Computational Linguistics: Human Language Technologies. 2021.
>
> Cohen, William, Vitor Carvalho, and Tom Mitchell. "Learning to classify email into “speech acts”." Proceedings of the 2004 Conference on Empirical Methods in Natural Language Processing. 2004.

---

### Meta-Review · Area_Chair_F4E1 · 2023-09-26

**Recommendation:** 4

**Metareview:**

The paper presents a dataset for event extraction from a conversational email threads. The dataset could be a valuable contribution to the research community. The choice of event types is very narrow and in line with a very old literature. The intuition or criterion of the schema used is not well motivated and prompting should be described in a more comprehensible manner.

---

### Decision · Program_Chairs · 2023-10-07

**Decision:**

Accept-Main

**Comment:**

The paper presents a dataset for event extraction from a conversational email threads. The dataset could be a valuable contribution to the research community. The choice of event types is very narrow and in line with a very old literature. The intuition or criterion of the schema used is not well motivated and prompting should be described in a more comprehensible manner.